# OmniSapiens: A Foundation Model for Social Behavior Processing via Heterogeneity-Aware Relative Policy Optimization

**Keane Ong** [1 2]  **Sabri Boughorbel** [3]  **Luwei Xiao** [1]  **Chanakya Ekbote** [2]  **Wei Dai** [2]  **Ao Qu** [2]  **Jingyao Wu** [2]
**Rui Mao** [4]  **Ehsan Hoque** [5]  **Erik Cambria** [4]  **Gianmarco Mengaldo**[† 1]  **Paul Pu Liang**[† 2]

## Abstract

Socially intelligent AI systems must reason across diverse human behavioral tasks and generalize to new social contexts. However, behavioral data is inherently heterogeneous, comprising diverse modalities and prediction targets that produce uneven training signals across samples, creating imbalanced learning dynamics that challenge existing AI models. To address this, we develop OMNISAPIENS-7B 2.0, a foundation model for social behavior processing that explicitly addresses learning from heterogeneous behavioral data. This is enabled through HETEROGENEITY-AWARE RELATIVE POLICY OPTIMIZATION, a new RL method that rebalances learning signals across samples by approximating each sample's contribution to the policy update and using these estimates to drive geometrically centered, inertially smoothed advantage modulation for stable training. OMNISAPIENS-7B 2.0 achieves the best and most consistent performance across 10 behavioral tasks, while also attaining the best performance on all five held-out benchmarks, with gains of up to +12.02% and +9.37% respectively. Furthermore, it demonstrates more consistent and interpretable reasoning traces, supporting reliable real-world behavioral applications. Our model is available at https://github.com/MIT-MI/human_behavior_atlas.

## 1. Introduction

Social intelligence, the ability to interpret affective expressions, mental states, and social signals, does not emerge from an isolated understanding of behavioral cues (Baron-Cohen, 2001; Goleman, 2006). How can we infer intent without perceiving social context? How can we understand sarcasm without understanding humor? Social intelligence requires reasoning across interconnected behavioral contexts, where distinct signals interact to produce coherent understanding and adaptation to new situations (Kihlstrom & Cantor, 2000). Yet, existing AI systems are predominantly specialized for a single behavioral task (e.g., sentiment analysis), with limited progress toward unified models that can effectively generalize across diverse behavioral contexts.

To develop unified social behavioral models, recent work has curated large-scale behavioral benchmarks (Ong et al., 2026) and adopted reasoning-based RL (Zhang et al., 2025a) to elicit the structured reasoning behavioral tasks require. However, unified models still remain significantly constrained by the imbalanced learning dynamics induced from behavioral data. Behavioral data is highly heterogeneous, spanning distinct feature types (e.g., acoustic prosody, interpersonal interaction) and prediction targets (e.g., emotion classification (Zadeh et al., 2016), social inference (Wilf et al., 2023)), which often produces uneven training signals across samples. Existing unified models (Ong et al., 2026) typically employ RL methods, i.e., GRPO (Shao et al., 2024), to enhance reasoning performance. However, these methods aggregate policy updates across heterogeneous samples without explicitly regulating disparities in their learning dynamics. As a result, optimization can become disproportionately dominated by specific behavioral tasks or samples, hindering multitask performance and cross-context generalization.

To address these challenges, we build OMNISAPIENS-7B 2.0, a foundation model for social behavior processing across 10 behavioral tasks. OMNISAPIENS-7B 2.0 is trained with HETEROGENEITY-AWARE RELATIVE POLICY OPTIMIZATION (HARPO), an on-policy optimization mechanism that rebalances learning across heterogeneous behavioral data to ensure that no single task or sample disproportionately influences policy optimization. This is achieved by approximating contribution signals to the policy update, and using them to inform geometrically centered and inertially smoothed advantage modulation.

---

[†]Equal advising. [1]National University of Singapore [2]Massachusetts Institute of Technology [3]Prince Sattam bin Abdulaziz University [4]Nanyang Technological University [5]University of Rochester. Correspondence to: Keane Ong <keaneong@mit.edu>.

*Proceedings of the 43rd International Conference on Machine Learning*, Seoul, South Korea. PMLR 306, 2026. Copyright 2026 by the author(s).

Across 10 behavioral tasks spanning diverse cognitive, affective, pathological, and social phenomena, OMNISAPIENS-7B 2.0 achieves significantly stronger multitask performance than existing behavioral models, with improvements of up to +12.02%. Beyond multitask learning, OMNISAPIENS-7B 2.0 further demonstrates the strongest zero-shot generalization to five held-out behavioral benchmarks spanning clinical behavior, affective expression, and social-pragmatic reasoning, outperforming prior models by up to +9.37%. Beyond predictive performance, OMNISAPIENS-7B 2.0 produces more interpretable reasoning traces, improving key XAI interpretability desiderata including robustness, efficiency, understandability, completeness, and compactness. Finally, compared against recent critic-free reasoning RL methods, HARPO achieves the most consistently strong performance across behavioral tasks, outperforming GRPO by up to +42.29%.

Our contributions are as follows: (1) We develop and release OMNISAPIENS-7B 2.0, a foundation model for unified social behavior analysis. OMNISAPIENS-7B 2.0 is one of the first models to jointly tackle a wide range of behavioral analysis tasks spanning affective, cognitive, pathological, and social dimensions. OMNISAPIENS-7B 2.0 significantly outperforms prior state-of-the-art models across 10 diverse behavioral tasks, while also demonstrating substantially stronger generalization in novel behavioral settings. Furthermore, OMNISAPIENS-7B 2.0 produces more interpretable reasoning traces, improving transparency and reliability for real-world behavioral analysis applications. Together, these capabilities unlock potential applications in affective computing, behavioral health, and socially intelligent AI systems, particularly in settings where behavioral data is scarce and model interpretability is critical; (2) we address the challenges associated with learning from heterogeneous behavioral data by introducing HARPO, a critic-free, reasoning-based RL method designed to learn from heterogeneous behavioral data.

## 2. Related Work

**Social behavior processing** develops AI for modelling human behavioral markers. Existing studies emphasize task-specific modeling, including interpreting *affective states* via emotion and arousal–valence classification (Zadeh et al., 2016; Dang et al., 2023); *cognitive states* through stress and cognitive load estimation (Giannakakis et al., 2019); *pathological states* by detecting depression and anxiety (Joshi & Kanoongo, 2022; Miloyan et al., 2014); and *social processes* through humor and engagement detection (Hessel et al., 2023; Monkaresi et al., 2016). While useful, task-specific processing overlooks the interdependencies between behavioral dimensions (Pessoa, 2008), motivating unified approaches for representation sharing (Ong et al., 2026).

**Reasoning-based reinforcement learning** has included recent advances such as GRPO (Shao et al., 2024) for group-normalized rewards, REINFORCE Leave-One-Out (Ahmadian et al., 2024) which reduces gradient variance, REINFORCE++ (Hu, 2025) which improves optimization stability, and Group Policy Gradient (Chu et al., 2025) which models group-level objectives. Yet, reasoning RL for learning heterogeneous behavioral data remains underexplored.

**Multitask learning** has evolved from unimodal representation sharing to any-to-any architectures that jointly model multiple modalities (Liang et al., 2024). Prior work has also explored gradient balancing (Yu et al., 2020), uncertainty-based weighting (Kendall et al., 2018), shared distilled policies (Teh et al., 2017). Yet, multitask learning remains largely underexplored for recent reasoning RL.

## 3. OmniSapiens-7B 2.0

### 3.1. Model Setup

**Multimodal backbone.** Given the multimodal nature of the behavioral data (text, vision, and audio) (Liang et al., 2024), we adopt Qwen 2.5 Omni-7B (Xu et al., 2025a) as our base architecture for OMNISAPIENS-7B 2.0. All training follows a multitask setup, where a single model is jointly trained across all tasks.

**Behavioral training data.** We use the Human Behavior Atlas benchmark (Ong et al., 2026) for training, which covers diverse human behavior tasks over 100k samples. These include sentiment polarity (SEN), emotion recognition (EMO), social reasoning (SOC), intent recognition (INT), non-verbal communication (NVC), as well as detecting humor (HUM), sarcasm (SAR), anxiety (ANX), depression (DEP), and PTSD (PTSD). For training, we develop HARPO, a reasoning-RL algorithm that explicitly balances heterogeneous learning signals across behavioral tasks.

### 3.2. Preliminaries

**Problem: Learning multiple behavioral tasks.** We consider the problem of training a stochastic policy $\pi_\theta(a \mid s)$, instantiated as a large language model (LLM), to perform a diverse set of behavioral understanding tasks indexed by $m \in \mathcal{M}$. For each task $m$, a sample $q \sim \mathcal{D}_m$, which can include multimodal inputs (e.g., text, audio, or visual signals), is drawn from a task-specific input distribution, and the policy generates an autoregressive output sequence $o \sim \pi_\theta(\cdot \mid q)$, with tokens $o_{:k}$ and prefixes $o_{:<k}$. The learning objective is to train a single shared policy $\pi_\theta$ that maximizes performance across all tasks $m \in \mathcal{M}$.

**Group Relative Policy Optimization for behavioral tasks.** To optimize $\pi_\theta(a \mid s)$, we consider Group Relative Policy Optimization (GRPO) (Shao et al., 2024), a recent

on-policy RL method that has shown strong performance for reasoning-based LLM training. For task $m$ and sample $q$, GRPO samples a rollout group $G_{(m,q)}$ of responses $\{o_{(m,q,i)}\}$, where $i \in G_{(m,q)}$ indexes individual rollouts (i.e., a sampled response) with rewards $r_{(m,q,i)}$, computing the group-normalized advantage:

$$\hat{A}_{(m,q,i)} = \frac{r_{(m,q,i)} - \hat{\mu}_{G_{(m,q)}}}{\hat{\sigma}_{G_{(m,q)}} + \varepsilon} \tag{1}$$

where $\hat{\mu}_{G_{(m,q)}}$ and $\hat{\sigma}_{G_{(m,q)}}$ are the mean and standard deviation of $\{r_{(m,q,i)}\}_{i=1}^{|G_{(m,q)}|}$. With a PPO clipped surrogate $\tilde{A}_{(m,q,i):k}(\theta)$ constructed from $\hat{A}_{(m,q,i)}$, GRPO then optimizes $\pi_\theta(a \mid s)$ using a PPO-style trust-region objective[1]:

$$J_{\mathrm{GRPO}}(\theta) = \mathbb{E}_{(m,q)\sim\mathcal{D}}\mathbb{E}_{\{o_{(m,q,i)}\}\sim\pi_{\theta_{\mathrm{old}}}}\left[\frac{1}{|G_{(m,q)}|}\sum_{i\in G_{(m,q)}}\right.$$
$$\left.\frac{1}{n_{o_{(m,q,i)}}}\sum_{k=1}^{n_{o_{(m,q,i)}}}\tilde{A}_{(m,q,i):k}(\theta)\right] - \beta\,\mathbb{E}[D_{\mathrm{KL}}(\pi_\theta \parallel \pi_{\mathrm{ref}})] \tag{2}$$

By the policy-gradient theorem (Sutton et al., 1999), the policy gradient admits the standard form, where the expectation is over $(s, a)$ induced by $\pi_\theta$:

$$\nabla_\theta J(\theta) = \mathbb{E}[A^{\pi_\theta}(s, a)\,\nabla_\theta \log \pi_\theta(a \mid s)] \tag{3}$$

When GRPO optimizes a policy across diverse behavioral tasks, gradient contributions from different $m$, $q$, and $i$ are aggregated into a shared update. Accordingly, a Monte Carlo estimator of the policy gradient, $g(\theta)$, can be decomposed as:

$$g(\theta) = \sum_{m\in\mathcal{M}}\sum_{q\sim\mathcal{D}_m}\sum_{i\in G_{(m,q)}} g_{(m,q,i)}(\theta) \tag{4}$$

Each rollout contributes a gradient term of the form:

$$g_{(m,q,i)}(\theta) \triangleq \hat{A}_{(m,q,i)}\,\nabla_\theta \log \pi_\theta(a_{(m,q,i)} \mid s_{(m,q,i)}) \tag{5}$$

From Eq.(5), each rollout contributes a gradient term whose magnitude is scaled by its advantage value. Since these rollout-level gradients are aggregated to form the shared policy gradient in Eq.(4), the resulting policy update is sensitive to the scale of advantages across rollouts. Across diverse behavioral tasks $m$, reward and advantage distributions may vary considerably because the associated behavioral data are highly heterogeneous, spanning different multimodal features (e.g., acoustic prosody, facial expressions) and prediction targets (e.g., emotion classification, social inference). Consequently, rollouts from tasks or samples with systematically elevated or suppressed advantage magnitudes can

exert disproportionate influence on GRPO's policy update, contributing to uneven learning across tasks (Sec. 4.4 provides an empirical illustration; App. F.3 Fig. 9 shows the differences in task advantage distributions).

### 3.3. Heterogeneity-Aware Relative Policy Optimization

**Dynamic advantage modulation mechanism.** To address this gap, we introduce HETEROGENEITY-AWARE RELATIVE POLICY OPTIMIZATION (HARPO). Since rollout influence on the policy update scales with advantage magnitude, Eq. (4), HARPO's core insight is to modulate advantages such that no single task or sample disproportionately influences policy optimization. In practice, our modulation mechanism scales the GRPO group-normalized advantages, Eq. (1), to mitigate imbalances in advantages at two levels: the sample-level, corresponding to advantages in a sample's rollout group, and the task-level, corresponding to advantages across all rollout groups of a task. This scaling is updated dynamically across training, using estimates of the relative contribution of each sample and task to the policy update. Concretely, we first construct a contribution signal $p^{(t)}$ from advantages, which approximates each task's and sample's relative contribution to the shared policy update. These signals are then transformed into structured, geometrically centered modulation factors, $\{s_{(m,q)}^{(t)}, \ s_m^{(t)}\}$, that scale the advantages prior to the policy update step. Finally, the modulation factors are updated using inertial smoothing to ensure their stability.

**Contribution signals from advantage magnitudes.** We construct a proxy signal, $p^{(t)}$, to approximate contribution strength to the policy update. Since policy gradients are directly scaled by advantages, Eq.(4)-(5), advantage magnitudes provide a computationally convenient proxy for contribution strength. Therefore, we use advantage magnitudes, normalized by rollout count to ensure invariance to stochastic batch sampling (Schulman et al., 2017), and use this to define $p^{(t)}$ at two levels. While the sample-level $p_{(m,q)}^{(t)}$ approximates the contribution of sample $q$ within a task $m$, the task-level $p_m^{(t)}$ approximates the contribution of a task $m$. Accordingly, $p_{(m,q)}^{(t)}$ is the average absolute group-normalized advantage over a rollout group $G_{(m,q)}$ corresponding to a sample $q$ of a task $m$ at training step $t$.

$$p_{(m,q)}^{(t)} = \frac{1}{|G_{(m,q)}|}\sum_{i\in G_{(m,q)}}|\hat{A}_{(m,q,i)}^{(t)}| \tag{6}$$

Then, we define $p_m^{(t)}$ as the average absolute group-normalized advantage over all rollouts collected for task $m$, with $\mathcal{Q}_m^{(t)}$ denoting the set of samples drawn for task $m$:

$$p_m^{(t)} = \frac{\sum_{q\in\mathcal{Q}_m^{(t)}}\sum_{i\in G_{(m,q)}}|\hat{A}_{(m,q,i)}^{(t)}|}{\sum_{q\in\mathcal{Q}_m^{(t)}}|G_{(m,q)}|} \tag{7}$$

---

[1]For completeness, the full formulation of GRPO is provided in App. E.1.

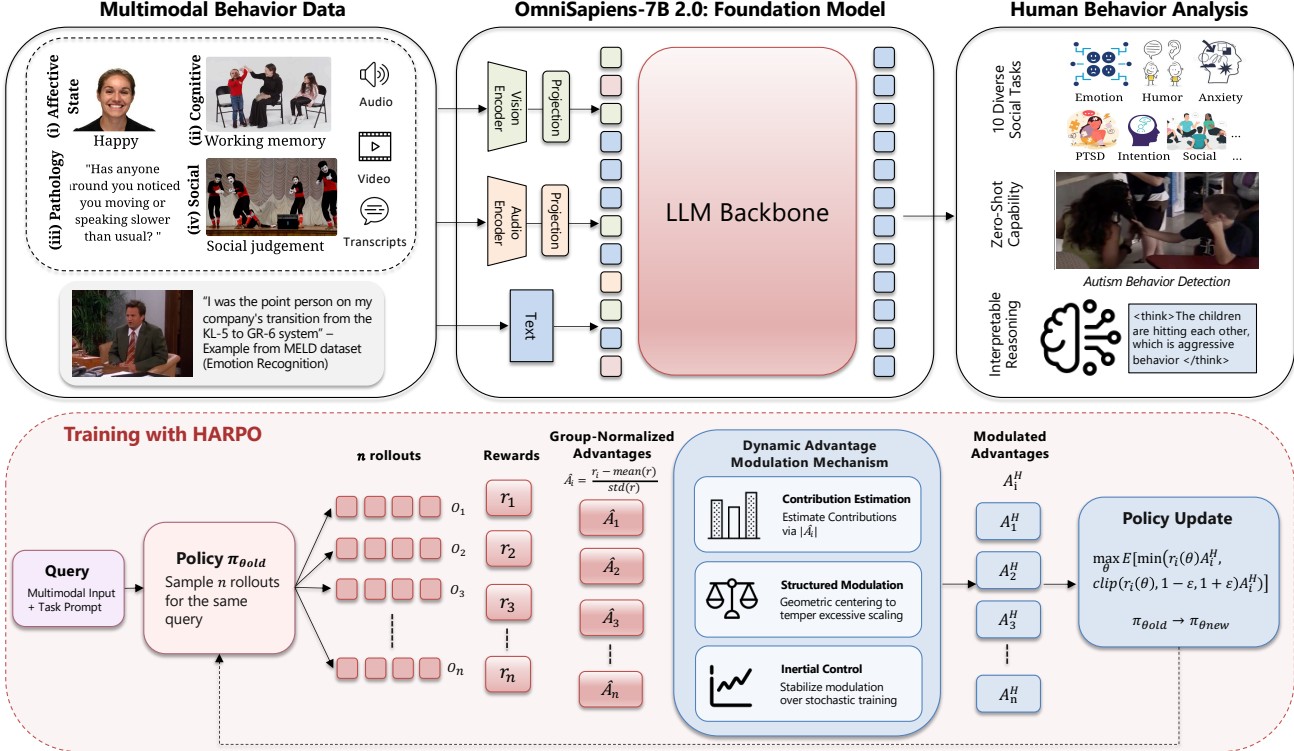

*Figure 1.* Model overview. OMNISAPIENS-7B 2.0 significantly outperforms prior state-of-the-art models across 10 diverse behavioral tasks and demonstrates substantially stronger generalization in novel behavioral settings. In addition, OMNISAPIENS-7B 2.0 produces relatively more interpretable reasoning traces, improving transparency and reliability for real-world behavioral analysis applications.

**Structured modulation via a geometric reference.** We leverage contribution signals to rebalance influence on the policy update, amplifying samples and tasks with lower contribution signals, and attenuating those with higher ones. Accordingly, we measure how far the contribution signal of each sample and task lay above or below a geometric-mean reference, and proportionately downscale or upscale their associated advantages. On the one hand, taking reference from the geometric-mean ensures that scaling is directly comparable across samples and tasks, and does not change the overall update size. On the other hand, it yields a ratio-based construction of scaling factors that tempers excessive variations in values (as we observe that contribution signals can vary by orders of magnitude, App. F.3 Fig. 8).

Concretely, we define a geometric mean reference at both the sample-level, $\bar{p}_{\text{ref},m}^{(t)}$, and the task-level, $\bar{p}_{\text{ref},\mathcal{M}}^{(t)}$:

$$\bar{p}_{\text{ref},m}^{(t)} = \left( \prod_{q \in \mathcal{Q}_m^{(t)}} p_{(m,q)}^{(t)} \right)^{\frac{1}{|\mathcal{Q}_m^{(t)}|}} \quad \bar{p}_{\text{ref},\mathcal{M}}^{(t)} = \left( \prod_{m \in \mathcal{M}} p_m^{(t)} \right)^{\frac{1}{|\mathcal{M}|}} \tag{8}$$

This establishes common baselines for comparing contributions across samples within a task or across tasks overall. For each sample or task contribution signal, we take the

reciprocal of its ratio to the geometric-mean. This yields modulation factors $s_{(m,q)}^{(t)}$ at the sample-level, and $s_m^{(t)}$ at the task-level.

$$s_{(m,q)}^{(t)} = \frac{\bar{p}_{\text{ref},m}^{(t)}}{p_{(m,q)}^{(t)}} \qquad s_m^{(t)} = \frac{\bar{p}_{\text{ref},\mathcal{M}}^{(t)}}{p_m^{(t)}} \tag{9}$$

The factors scale the group-normalized advantages of each rollout, yielding the HARPO advantage:

$$A_{(m,q,i)}^{\text{H}\,(t)} \triangleq s_{(m,q)}^{(t)} \, s_m^{(t)} \, \hat{A}_{(m,q,i)}^{(t)} \tag{10}$$

Due to their reciprocal construction, the modulation factors act to balance advantage magnitudes across samples and tasks. Specifically, within a task, samples whose contribution signals exceed the sample reference (i.e., $p_{(m,q)}^{(t)} > \bar{p}_{\text{ref},m}^{(t)}$) receive modulation factors $s_{(m,q)}^{(t)} < 1$, while those below the reference (i.e., $p_{(m,q)}^{(t)} < \bar{p}_{\text{ref},m}^{(t)}$) receive $s_{(m,q)}^{(t)} > 1$. Consequently, within the same task, stronger samples have downscaled advantages while weaker samples have upscaled advantages. Analogously, at the task-level, the advantages of tasks with contribution signals above the reference $\bar{p}_{\text{ref},\mathcal{M}}^{(t)}$ are downscaled, while those below the reference are upscaled. Additionally, as the modulation factors are constructed from a geometric mean reference, the fac-

tors have a geometric mean of 1. ($\prod_{q \in \mathcal{Q}_m^{(t)}} s_{(m,q)}^{(t)} = 1$ and $\prod_{m \in \mathcal{M}} s_m^{(t)} = 1$, App. E.2 shows the full derivation). This ensures that multiplicative upscaling from certain modulation factors are exactly compensated by downscaling from others. Thus, the factors cannot simultaneously enlarge or shrink all advantages at the sample or task-level, mitigating unintended influence on the global step size.

**Inertial control for stable modulation.** The modulation mechanism is updated on a slower time scale than the policy parameters. Accordingly, we maintain inertial estimates of the contribution signals and the modulation factors, allowing modulation to respond to persistent trends in contribution signals, rather than stochastic on-policy fluctuations. Contribution signals are smoothed using an exponential moving average to mitigate stochastic noise (Kingma & Ba, 2014). Modulation factors, as multiplicative ratios, are smoothed via multiplicative updates rather than by additive adjustments (Arora et al., 2012; Bubeck et al., 2015). For notational simplicity, we use $p^{(t)}$ and $s^{(t)}$ to denote both the newly computed quantities from the current rollout batch and their recursively updated running estimates, with the latter serving as the operational quantities used in modulation.

$$
\begin{aligned}
p_{(m,q)}^{(t)} &= \beta_\rho \, p_{(m,q)}^{(t-1)} + (1 - \beta_\rho) \, p_{(m,q)}^{(t)} \\
p_m^{(t)} &= \beta_\rho \, p_m^{(t-1)} + (1 - \beta_\rho) \, p_m^{(t)} \\
s_{(m,q)}^{(t)} &= \left( s_{(m,q)}^{(t-1)} \right)^{\beta_s} \left( s_{(m,q)}^{(t)} \right)^{1-\beta_s} \\
s_m^{(t)} &= \left( s_m^{(t-1)} \right)^{\beta_s} \left( s_m^{(t)} \right)^{1-\beta_s}
\end{aligned}
\tag{11}
$$

**HARPO[2] objective.** With the HARPO-modulated advantage in place, HARPO retains the PPO-style trust-region objective of GRPO, substituting $A^{\mathrm{H}}$ for $\hat{A}$ to construct the clipped surrogate $\tilde{A}_{(m,q,i):k}^{\mathrm{H}}(\theta)$, thereby forming the HARPO objective:

$$
J_{\mathrm{HARPO}}(\theta) = \mathbb{E}_{(m,q)\sim\mathcal{D}} \mathbb{E}_{\{o_{(m,q,i)}\}\sim\pi_{\theta_{\mathrm{old}}}} \left[ \frac{1}{|G_{(m,q)}|} \sum_{i \in G_{(m,q)}} \right.
$$

$$
\left. \frac{1}{n_{o_{(m,q,i)}}} \sum_{k=1}^{n_{o_{(m,q,i)}}} \tilde{A}_{(m,q,i):k}^{\mathrm{H}}(\theta) \right] - \beta \, \mathbb{E}[D_{\mathrm{KL}}(\pi_\theta \| \pi_{\mathrm{ref}})]
\tag{12}
$$

### 3.4. Reward Design

**Accuracy.** For classification tasks, we use a binary reward $r_{\mathrm{cls}}$, which equals 1 for an exact label match and 0 otherwise. For question-answering tasks with free-text responses, we use cosine similarity rewards, $r_{qa}$, normalized to a scale of $[0, 1]$ for compatibility with rewards assignment, to measure alignment between the generated response and answer.

---

[2]App. A Algorithm 1 summarizes HARPO. Additional details of reward design are in App. C.2

**Formatting.** We add a binary reward, $r_{\mathrm{fmt}}$, for adhering to the response structure, which includes reasoning traces followed by the model's prediction.

**Length.** Following Zhang & Zuo (2025), we leverage an overlong length penalty, $r_{\mathrm{len}}$, to prevent excessive verbosity of responses.

**Final reward[2].** With format weight $w_{\mathrm{fmt}} = 0.2$ and length scale $\lambda_{\mathrm{len}} = 0.75$, the final per-sample reward is the following, where $r_{task}$ can be $r_{cls}$ or $r_{qa}$, depending on if the sample belongs to a classification or QA task respectively.

$$
r = (1 - w_{\mathrm{fmt}}) \, r_{\mathrm{task}} + w_{\mathrm{fmt}} \, r_{\mathrm{fmt}} + \lambda_{\mathrm{len}} \, r_{\mathrm{len}},
$$

## 4. Experiments

Leveraging the Human Behavior Atlas benchmark (Ong et al., 2026), we analyze the capabilities of OMNISAPIENS-7B 2.0 as a foundation model for social behavior processing. We conduct our evaluation across three key dimensions. First, we assess *multitask capability*, evaluating whether a single model can coherently execute diverse behavioral analysis tasks, thereby obviating the need to train, store, and deploy separate task-specific models. Second, we evaluate *zero-shot generalization*, assessing whether learned representations transfer to unseen behavioral contexts without task-specific supervision. Such transfer mitigates annotation costs that typically constrain deployment to novel domains. Third, we analyze the *reasoning trace interpretability*, which crucially enables post-hoc verification of model judgements in critical behavioral applications (i.e., health).

### 4.1. Multitask Performance

RQ1: We evaluate OMNISAPIENS-7B 2.0 against: (1) general-purpose foundation model families: Gemma (Team et al., 2025), Qwen (Xu et al., 2025b); and (2) specialized models trained on human behavior data, comprising HumanOmniV2 (Yang et al., 2025) and prior Omnisapiens variants (SFT, BAM, RL) (Ong et al., 2026). Using the same model backbone, Qwen 2.5-Omni (Xu et al., 2025a), we also compare HARPO against current critic-free RL methods that explicitly address learning heterogeneous data, (1) normalization-based: GRPO (Shao et al., 2024), RE++ (Hu, 2025), (2) variance-reduction: RLOO (Ahmadian et al., 2024), (3) unbiased gradients: GPG (Chu et al., 2025), (4) multitask balancing: EMAGRPO (Feng et al., 2025).

**Comparing model performance**. OMNISAPIENS-7B 2.0 achieves state-of-the-art performance for multitask social behavior processing. Relative to existing models, OMNISAPIENS-7B 2.0 achieves the strongest overall performance across multiple behavioral tasks. From Tab. 1, OMNISAPIENS-7B 2.0 outperforms general-purpose LLMs (Gemma-3, Gemma-4, Qwen-2.5-Omni, Qwen-2.5-VL,

*Table 1.* Per-task performance (%) is reported for both models and training algorithms. Each task may include multiple datasets; we report the mean performance across a task's associated datasets, with full results in App. Tab. 9. Following Ong et al. (2026), we report binary weighted F1 for SEN; mean per-class weighted accuracy for EMO; weighted F1 for HUM, SAR, ANX, DEP, and PTSD; and LLM-Judge accuracy for SOC, INT, and NVC. Given the disparities between the metrics, we compute average performance rank across tasks separately for models and algorithms (↓ indicates lower score is better), with ties assigned the same rank. Best results are bolded and second-best are underlined. (*) denotes existing models that have been trained on more than two behavioral tasks, reflecting substantive multitask training; only a few such models are publicly available. For fair comparison, all training algorithm results reflect training on the same Human Behavior Atlas benchmark, using an identical reward design (Sec. 3.4) and the same base model (Qwen2.5-Omni-7B).

| Models | EMO | HUM | INT | PTSD | ANX | DEP | SEN | SAR | SOC | NVC | Avg. Rank ↓ |
|---|---|---|---|---|---|---|---|---|---|---|---|
| Gemma 4-E4B (8B) | 55.98 | 52.63 | 40.25 | 86.00 | 33.36 | 22.55 | 69.55 | 73.54 | 20.40 | 4.65 | 7.30 |
| Gemma-3-4B | 55.03 | 59.70 | 22.70 | 49.90 | 60.10 | 46.25 | 73.83 | 52.90 | 19.10 | 2.30 | 8.30 |
| Qwen 2.5-Omni-7B | 58.25 | 54.30 | 25.40 | 76.00 | 79.30 | 71.35 | 67.20 | 65.60 | 25.40 | 6.90 | 6.20 |
| Qwen 2.5-VL-7B | 54.08 | 58.30 | 24.90 | 75.50 | 63.10 | 63.80 | 50.50 | 51.10 | 23.10 | 9.80 | 7.80 |
| Qwen 3-VL-8B-Instruct | 57.66 | 66.76 | 38.00 | 92.70 | 42.29 | 51.62 | 69.70 | 63.67 | 24.94 | 13.95 | 5.50 |
| HumanOmniV2-7B* | 59.70 | 63.80 | 26.30 | 82.40 | 52.70 | 65.40 | 74.20 | 39.50 | 28.20 | 9.30 | 5.80 |
| OmniSapiens SFT* | 63.08 | 53.20 | 25.60 | **100.00** | 90.90 | 73.25 | 76.77 | 62.40 | 25.70 | 12.10 | 4.40 |
| OmniSapiens BAM* | 64.53 | 64.40 | 17.70 | **100.00** | 90.90 | 78.85 | **78.53** | **79.50** | 20.10 | **16.20** | 3.30 |
| OmniSapiens-7B RL* | 57.28 | 63.90 | 48.60 | 96.80 | 91.90 | 77.15 | 39.60 | 64.70 | **30.40** | 13.30 | 4.20 |
| **OmniSapiens-7B 2.0 (ours)** | **76.55** | **69.85** | **50.52** | 98.39 | **91.98** | **78.87** | 77.61 | 70.64 | 25.40 | 14.54 | **1.90** |
| **Training Algorithms** | | | | | | | | | | | |
| RLOO (Ahmadian et al., 2024) | 75.58 | 67.86 | 51.73 | **98.39** | 90.68 | 77.57 | 76.86 | 62.58 | 29.54 | **16.28** | 2.80 |
| RE++ (Hu, 2025) | 75.92 | 60.26 | 5.01 | **98.39** | **93.11** | 73.87 | 56.52 | 50.21 | 12.64 | 4.07 | 4.50 |
| GPG (Chu et al., 2025) | **77.70** | 69.28 | **54.21** | **98.39** | 90.40 | 78.40 | 75.77 | 45.96 | 27.93 | 12.79 | 2.90 |
| GRPO (Shao et al., 2024) | 76.45 | 27.56 | 49.90 | **98.39** | 90.40 | 77.64 | 77.51 | 53.58 | 23.30 | 11.00 | 3.90 |
| EMAGRPO | 73.36 | 63.50 | 52.62 | **98.39** | 90.68 | 73.36 | 68.28 | **77.75** | **30.58** | 15.70 | 3.10 |
| **HARPO (ours)** | 76.55 | **69.85** | 50.52 | **98.39** | 91.98 | **78.87** | **77.61** | 70.64 | 25.40 | 14.54 | **2.10** |

Qwen-3-VL). It also outperforms prior behavioral foundation models for diverse social-behavioral tasks spanning multiple behavioral dimensions, including HumanOmniV2-7B and OmniSapiens-7B RL, BAM, SFT. In particular, OMNISAPIENS-7B 2.0 ranks in the top 2 on 8 of 10 tasks, with the strongest average task performance rank (1.90).

**RL algorithm performance comparison**. Among state-of-the-art reasoning RL algorithms, HARPO achieves the most consistent performance across behavioral tasks. As shown in Tab. 1, HARPO outperforms reasoning-based RL methods spanning different approaches, normalization: GRPO (Shao et al., 2024), RLOO (Ahmadian et al., 2024), variance reduction: RE++ (Hu, 2025), unbiased gradients: GPG (Chu et al., 2025), and multitask balancing: EMA-GRPO (Feng et al., 2025), attaining the best average rank (2.10) across tasks. While GRPO, GPG, and RE++ performance collapse on certain tasks (e.g., SAR), HARPO does not exhbit the same performance degradation on weaker tasks, indicating more balanced multitask learning.

### 4.2. Zero-Shot Generalization Performance

RQ2: We evaluate zero-shot generalization of OMNISAPIENS-7B 2.0 on five diverse held-out behavioral tasks, without further fine-tuning: (1) Autism Detection (AUT) (Deng et al., 2024); (2) Speech Emotion Recognition (SER) (Busso et al., 2008); (3) In-the-wild depression recognition (IDR) (Shen et al., 2022); (4) Social Media Sentiment Analysis (SMSA) (Niu et al., 2016); (5) Sarcasm Incongruity Recognition (SIR) (Yue et al., 2024).

*Table 2.* Zero-shot generalization weighted F1 performance (%) on heldout tasks AUT, SER, IDR, SMSA, and SIR. *OmniSapiens-7B SFT and BAM are excluded, as their task-specific classifier heads structurally preclude zero-shot inference.

| Model | AUT | SER | IDR | SMSA | SIR |
|---|---|---|---|---|---|
| Qwen 2.5-Omni-7B | 25.68 | 53.53 | 70.25 | 44.64 | 34.99 |
| Gemma 4-E4B-Instruct (8B) | 16.78 | 58.30 | 11.37 | 55.60 | 56.74 |
| OmniSapiens-7B RL | 30.46 | 55.77 | 69.29 | 55.03 | 66.53 |
| HumanOmniV2-7B | 38.05 | 62.74 | 21.97 | 53.06 | 37.45 |
| **OmniSapiens-7B 2.0 (ours)** | **39.91** | **72.11** | **72.43** | **58.47** | **69.27** |

**Zero-shot generalization performance.** From Tab. 2, OMNISAPIENS-7B 2.0 achieves the best performance across all five tasks spanning a broad spectrum of behavioral contexts, including clinical behavior (AUT, IDR), affective expression (SER), and social-pragmatic reasoning (SMSA, SIR). The uniform gains across held-out tasks indicate that relative to existing models, OMNISAPIENS-7B 2.0 learns behavioral representations that can more effectively generalize and transfer beyond training datasets and distributions. As a result, OMNISAPIENS-7B 2.0 demonstrates stronger effectiveness than existing models when applied to new datasets and distributions, as frequently encountered in real-world deployment settings.

**Comparison with OmniSapiens-7B RL**. Despite being trained on the same Human Behavioral Atlas benchmark, OMNISAPIENS-7B 2.0, trained with HARPO instead of the standard GRPO, exhibits stronger generalization performance than OmniSapiens-7B RL across each held-out task. This coincides with improved multitask performance achieved by OMNISAPIENS-7B 2.0 from Tab. 1, with an average task rank of 1.90 compared to 4.20 for OmniSapiens-

7B RL. This suggests that more uniform learning across behavioral tasks through HARPO may promote the acquisition of transferable behavioral features, yielding improved generalization. Though we leave a more rigorous investigation of this relationship to future work.

### 4.3. Reasoning Analysis

RQ3: We analyze the interpretability of OMNISAPIENS-7B 2.0's reasoning traces[3], using the held-out datasets in Sec. 4.2 to evaluate under realistic deployment conditions, and to control for training data overlap. We evaluate interpretability through a multi-dimensional notion of explanation quality, grounded in established XAI desiderata (Langer et al., 2021; Nauta et al., 2023; Turbé et al., 2023; Wei et al., 2024; Ong et al., 2025). We capture properties *robustness*, *efficiency* via automated metrics, consistency and mean token length); properties *understandability*, *completeness*, *compactness* via human evaluation of coherence, specificity, and concision.

*Table 3.* Reasoning evaluation results for robustness and efficiency of reasoning traces. Consistency is the fraction of inputs on which all 5 independently sampled reasoning traces produce the same answer. Average tokens is taken across the 5 sampled traces per input. Paraphrase consistency is the fraction of inputs on which the predicted answers agree across two semantically equivalent prompt variants. All metrics are averaged across all samples. (↑ higher score is better; ↓ lower score is better)

| Model | Consistency ↑ | Paraphrase ↑ | Avg. Tokens ↓ |
|---|---|---|---|
| Qwen2.5-Omni-7B | 34.0 | 58.2 | 73.66 |
| Gemma 4-E4B (8B) | 66.5 | 79.2 | 211.10 |
| HumanOmniV2 | 50.0 | 78.2 | 195.90 |
| OmniSapiens-7B-RL | 55.1 | 80.7 | 57.69 |
| OmniSapiens 2.0 (ours) | **87.7** | **88.4** | **19.86** |

**Robustness and efficiency of reasoning.** Following prior work (Wang et al., 2022; Jie et al., 2024), robustness is assessed via two consistency measures: agreement across independently sampled reasoning traces (Consistency) and agreement across semantically equivalent prompt variants (Paraphrase). From Tab. 3, OMNISAPIENS-7B 2.0 achieves the strongest reasoning consistency (87.7 vs. 66.5 next best) and the highest paraphrase consistency (88.4 vs. 80.7 next best), indicating that its reasoning traces are both stable under stochastic sampling and robust to input variation. This consistency yields a more analyzable reasoning process, strengthening interpretability (Alvarez Melis & Jaakkola, 2018). Efficiency, measured by reasoning trace length (Avg. Tokens), reflects the cost of eliciting interpretability. OMNISAPIENS-7B 2.0 produces more concise traces (19.86 tokens vs. 57.69 next best), reducing inference cost and the amount of information the reader must process.

---

[3]We make public the reasoning traces, and provide the experimental setup for the consistency and paraphrase analysis, and human evaluation instructions, annotators in App. D.

*Table 4.* Human evaluation of OMNISAPIENS-7B 2.0 reasoning trace interpretability over baselines. Evaluation criteria is based on specificity, coherence and concision of reasoning traces. Tie+Win: % tied or preferred over baseline; Tie: % equal preference; Win[†]: % Win rate excluding ties. OMNISAPIENS-7B 2.0 preferred over, or at the minimum tied with, every baseline across all three criteria.

| vs. Baselines | Specificity | | | Coherence | | | Concision | | |
|---|---|---|---|---|---|---|---|---|---|
| | Tie+Win | Tie | Win[†] | Tie+Win | Tie | Win[†] | Tie+Win | Tie | Win[†] |
| Qwen2.5-Omni-7B | 80.6 | 35.3 | 66.0 | 90.0 | 24.0 | 85.3 | 99.3 | 13.3 | 99.3 |
| Gemma 4-E4B (8B) | 76.0 | 16.7 | 70.0 | 88.0 | 12.0 | 85.4 | 98.7 | 0.7 | 98.7 |
| HumanOmniV2 | 76.7 | 14.0 | 70.4 | 87.3 | 9.3 | 85.7 | 100.0 | 0.0 | 100.0 |
| OmniSapiens-7B-RL | 82.5 | 40.6 | 66.1 | 87.9 | 20.9 | 83.2 | 98.6 | 8.0 | 98.6 |
| Avg. Rates | 78.9 | 26.6 | 68.5 | 88.3 | 16.5 | 85.1 | 99.2 | 5.5 | 99.2 |

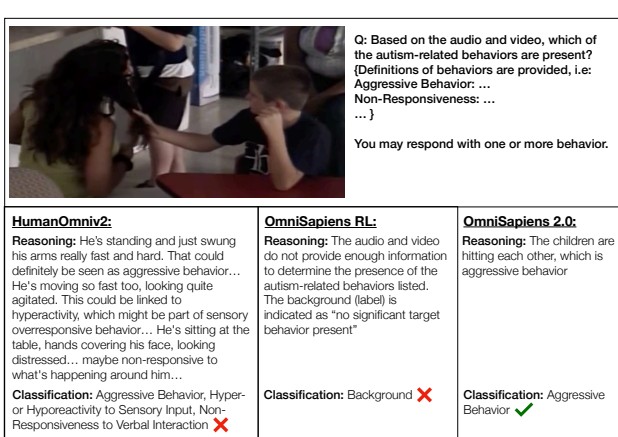

*Figure 2.* Reasoning case study for autism behavioral detection (AUT); HumanOmniv2 produces verbose reasoning, OmniSapiens RL abstains from predictions, OMNISAPIENS-7B 2.0 generates concise reasoning traces identifying the most salient behavior.

**Human evaluation and case study of the completeness, understandability and compactness of reasoning.** To assess reasoning quality, annotators performed pairwise comparisons between OMNISAPIENS-7B 2.0 and each baseline along three XAI desiderata: specificity (completeness), coherence (understandability), and concision (compactness). Averaged across baselines (Tab. 4), OMNISAPIENS-7B 2.0 is tied with or preferred over baselines in 78.9%, 88.3%, and 99.2% of comparisons for specificity, coherence, and concision respectively; excluding ties yields effective win rates of 68.5%, 85.1%, and 99.2%, indicating consistent preference for OMNISAPIENS-7B 2.0's more informative, logically structured, and succinct reasoning. A case study on the AUT task (Fig. 2) against existing behavioral foundation models further illustrates these trends: HumanOmniV2 generates expansive chains that introduce behaviors without clear evidential support, while OmniSapiens-7B-RL exhibits the opposite failure mode, defaulting to "background (no behavior)" predictions even amid salient cues. OMNISAPIENS-7B 2.0, by contrast, produces concise traces focused on cues directly supported by the clip, neither over-predicting nor over-abstaining, consistent with the specificity, coherence, and concision preferences observed in the human evaluation.

## 4.4. Empirical Analysis & Ablations

RQ4: To further evaluate HARPO, we conduct additional studies to examine its underlying mechanisms.

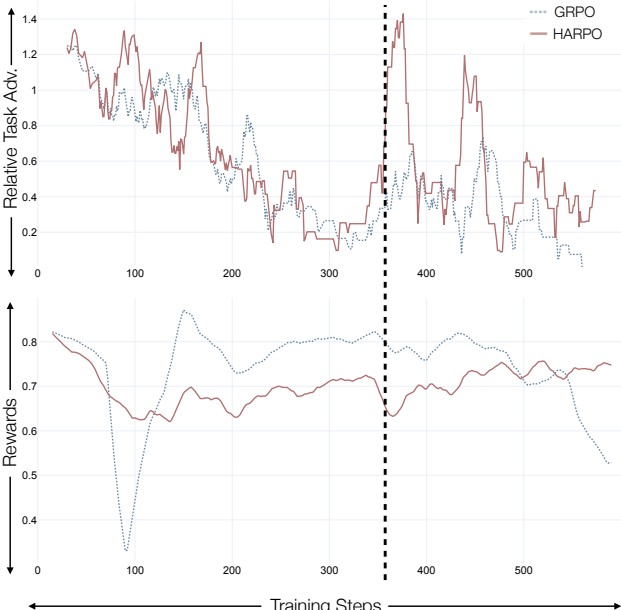

*Figure 3.* Comparison of relative task advantage and rewards for SAR. After step 355 (dotted line), HARPO exhibits increasing relative task advantage alongside increasing rewards, while GRPO maintains lower relative task advantage with decreasing rewards.

**HARPO's advantage modulation on weak performing tasks.** HARPO balances optimization by modulating advantages prior to the policy update step. To investigate this, we compute relative task advantage, defined as the ratio between a task's average advantage magnitude and the mean of the average advantage magnitudes across all tasks, to observe how a change in this statistic affects learning. We focus on SAR as a representative weak-performing task (performance of 53.58% under GRPO; 70.64% under HARPO), to more clearly study advantage modulation in this regime. From Fig. 3, in the later parts of training (i.e., after the dotted line depicting step 355), HARPO's amplification of SAR's relative task advantage coincides with a continued increase in average rewards. At the same training stage, GRPO's unmodulated relative task advantage remains lower, corresponding with a steady decline in rewards. This suggests that HARPO's advantage modulation sustains longer learning for weak-performing tasks with otherwise diminished advantages, contributing to improved performance.

**Structured modulation.** We ablate the structured modulation via geometric reference, and instead calculate advantage modulation factors from the reciprocal of the contribution signals ($s^{(t)} = 1/p^{(t)}$). From Tab. 5, this variant (w/o structured modulation) exhibits weaker multitask performance (avg. rank 2.00) compared to HARPO (1.90). An

*Table 5.* Average task performance rank for HARPO ablations (lower is better), computed by ranking methods per task based on performance, then averaging ranks across tasks. We provide the per-task and per-dataset performance breakdown in App. F.1.

| Variant | Avg. Rank ↓ |
|---|---|
| **HARPO** | **1.90** |
| w/o structured modulation | 2.00 |
| w/o sample-level modulation | 2.60 |
| w/o inertial control | 2.70 |

intuitive explanation is that the ablation produces a net amplification of advantages during training. The per-task modulation factors $s_m^{(t)}$ have a geometric mean that remains above 2.5 (App. Fig. 8), indicating sustained advantage upscaling during training. In contrast, HARPO maintains modulation factors with geometric mean 1 due to geometric centering, mitigating uniform amplification of advantages across tasks. Since policy-gradient is scaled by advantage values, increasing advantage magnitudes can alter the effective update size which adversely impacts optimization (Schulman et al., 2015; 2017). As a result, directly using $s^{(t)} = 1/p^{(t)}$ instead of a structured, geometrically centered modulation may inadvertently impact performance.

**Inertial control.** From Tab. 5, the average rank degrades from 1.90 with the HARPO method to 2.70 for the ablation without inertial control (i.e., without smoothing in Eq. 11). To study this further, we examine the task modulation factors $s_m^{(t)}$ over training for tasks SAR, SEN, ANX. Tasks with greater performance improvements with inertial control, SAR (+7.96%) and SEN (+5.69%) from App. F Tab. 7, also exhibit clearer differences in $s_m^{(t)}$ over training. From Fig. 4, inertial control yields gradual changes in $s_m^{(t)}$, in contrast to sharper fluctuations without it. In contrast, ANX, which exhibits comparatively smaller performance differences (+1.30%), shows similar $s_m^{(t)}$ trends with and without inertial control. This suggest that inertial control does not uniformly affect tasks, but can improve performance for tasks with excessive fluctuations in $s_m^{(t)}$, by stabilizing the scaling of advantage signals, which supports consistent policy updates (Ilyas et al., 2018).

**Modulation at the sample level.** To study modulating advantages at the sample-level in addition to the task-level, we run an ablation without sample-level modulation (i.e., leveraging only the task-level modulation factors, $s_m^{(t)}$). From Tab. 5, this results in weaker performance, with the ablation attaining a 2.60 avg. rank compared to HARPO at 1.90. To understand this, we analyze with Fig. 4, a task (ANX) that performs stronger with sample-level modulation. Accordingly, sample-level modulation produces a narrower advantage distribution than without, highlighting that relatively extreme advantage values are suppressed (we observe the same trend for other tasks NVC, SOC, HUM in App. F.3 Fig. 9). This narrower spread does not reduce aggregate

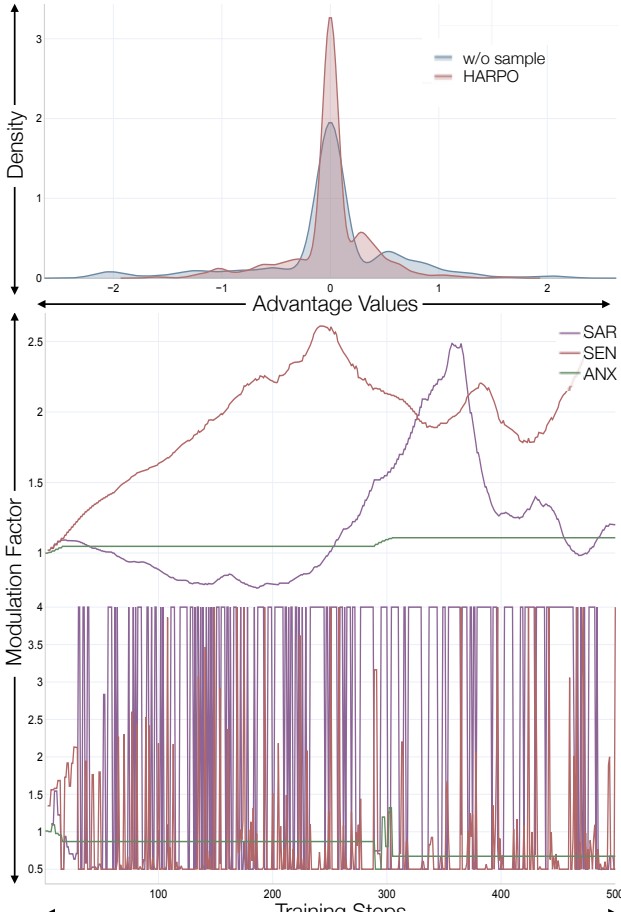

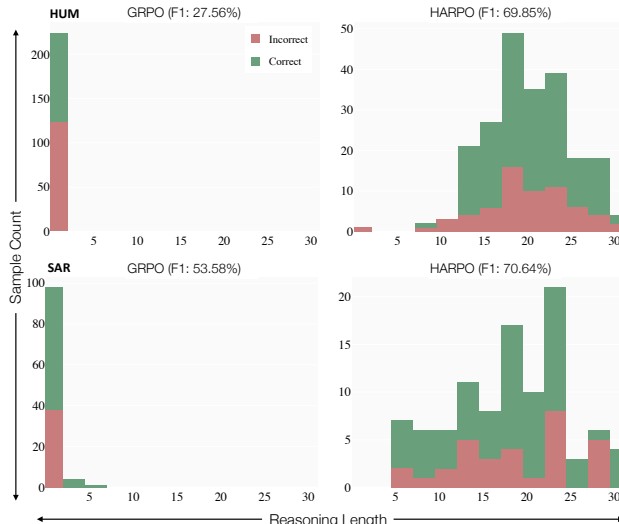

*Figure 5.* Sample count versus token reasoning length. Green indicates correct predictions, red indicates incorrect. HARPO induces more varied reasoning lengths for the respective tasks of (Top: HUM, Bottom: SAR), compared to GRPO.

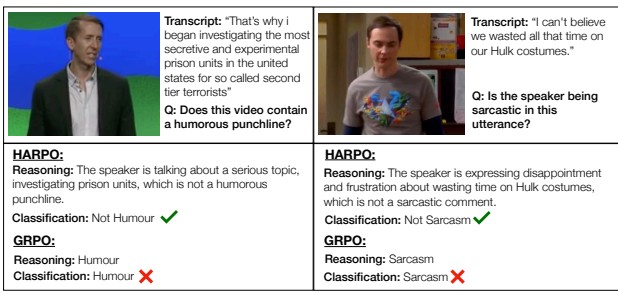

*Figure 6.* Example of reasoning traces on pragmatic tasks humor detection (HUM) (left) and sarcasm detection (SAR) (right). HARPO is observed to reflect more explicit and varied reasoning compared to GRPO, which defaults to minimal/ no reasoning.

*Figure 4.* Top: Comparison of distribution of advantages between HARPO and the ablation without sample-level modulation, for the ANX task. Sample-level modulation leads to a narrowing of the advantage distribution which can result in better performance for specific tasks (additional task distributions are in App. F). Middle & Bottom: Comparison of modulation factors between HARPO (middle) and without inertial control (bottom). Inertial control enables a more gradual change in modulation factor values.

advantages, since the geometric mean enforced by Eqs. (8)-(9) preserves the overall multiplicative scale of advantages. Instead, it prevents a subset of samples with extreme values from disproportionately influencing the policy update.

**HARPO's effect on reasoning traces**. For complex pragmatic tasks such as sarcasm (SAR) and humor (HUM) detection, which rely on pragmatic cues (i.e., contextual signals beyond literal lexical content), we observe that HARPO induces richer reasoning behavior as compared to GRPO, at inference. From Fig.5, HARPO achieves a larger proportion of correct predictions on SAR and HUM, compared to GRPO. This is also accompanied by longer and more varied reasoning-length distributions. Qualitative analysis, Fig. 6, further illustrates that GRPO often produces shortcut responses with minimal or no explicit reasoning, while HARPO's reasoning traces remain more contextually

grounded, reflecting deliberation over pragmatic and inferential cues present in the input. These findings suggest that HARPO reduces the suppression of useful reasoning trajectories for these weaker tasks, resulting in richer reasoning behavior that coincides with improved task performance.

## 5. Conclusion

In this work, we developed OMNISAPIENS-7B 2.0 a unified foundation model for social behavior analysis trained using HARPO, a new RL method that accounts for heterogeneous learning signals across multiple behavioral tasks. Multitask and generalization evaluations showed significantly improved performance across behavioral tasks and increased interpretability of reasoning traces, underscoring the value of modeling heterogeneous learning signals through HARPO. More broadly, these findings suggest how social behavioral foundation models may benefit from explicitly protecting weaker, but structurally meaningful signals that encode latent social and behavioral structure.

## Acknowledgements

This research/project is supported by the Asian Institute of Digital Finance, National University of Singapore; Ministry of Education, Singapore under its MOE Academic Research Fund Tier 2 (MOE-T2EP20123-0005: "Neurosymbolic AI for Commonsense-based Question Answering in Multiple Domains"), MOE Tier 1 Award (MOE-T2EP50221-0028: "Discipline-Informed Neural Networks for Interpretable Time-Series Discovery"), and by the RIE2025 Industry Alignment Fund – Industry Collaboration Projects (I2301E0026: "Generative AI"), administered by A*STAR, as well as supported by Alibaba Group and NTU Singapore. The authors would also like to acknowledge Modal and NVIDIA for their support through the provision of GPU and compute resources.

## Impact Statement

This paper presents work whose goal is to advance the field of Machine Learning. There are many potential societal consequences of our work, none of which we feel must be specifically highlighted here.

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

# A. HARPO Algorithm Block

We describe the full HARPO procedure in Algorithm 1. Unlike Eq. (11), where notation is collapsed for simplicity, the full algorithm explicitly distinguishes instantaneous contribution signals and modulation factors of the rollout batch, $\{p^{(t)}, s^{(t)}\}$, from their inertially controlled running estimates, $\{\tilde{p}^{(t)}, \tilde{s}^{(t)}\}$. The latter serves as the operational quantities used in modulation.

---

**Algorithm 1** HARPO: Heterogeneity Aware Relative Policy Optimization.

---

**Input:** task set $\mathcal{M}$; batch $\{(m,q)\} \sim \mathcal{D}$, where $m \in \mathcal{M}$ indexes tasks and $q$ denotes a sample from task $m$; rollout groups $\{G_{(m,q)}\}$ with rewards $\{r_{(m,q,i)}\}$; $\varepsilon > 0$; $\beta_\rho, \beta_s \in [0,1)$; previous inertial states $\tilde{p}^{(t-1)}$ and $\tilde{s}^{(t-1)}$

**Output:** updated policy parameters $\theta$

**Construct group-normalized advantages:**

**for all** $(m,q)$ in batch **do**

$\quad \hat{\mu}_{(m,q)} \leftarrow \frac{1}{|G_{(m,q)}|} \sum_{i \in G_{(m,q)}} r_{(m,q,i)}$

$\quad \hat{\sigma}_{(m,q)} \leftarrow \sqrt{\frac{1}{|G_{(m,q)}|} \sum_{i \in G_{(m,q)}} (r_{(m,q,i)} - \hat{\mu}_{(m,q)})^2}$

$\quad$ **for all** $i \in G_{(m,q)}$ **do**

$\quad\quad \hat{A}_{(m,q,i)}^{(t)} \leftarrow \frac{r_{(m,q,i)} - \hat{\mu}_{(m,q)}}{\hat{\sigma}_{(m,q)} + \varepsilon}$

$\quad$ **end for**

**end for**

**Construct contribution signals:**

**for all** $(m,q)$ in batch **do**

$\quad p_{(m,q)}^{(t)} \leftarrow \frac{1}{|G_{(m,q)}|} \sum_{i \in G_{(m,q)}} \left| \hat{A}_{(m,q,i)}^{(t)} \right|$

$\quad \tilde{p}_{(m,q)}^{(t)} \leftarrow \beta_\rho \, \tilde{p}_{(m,q)}^{(t-1)} + (1 - \beta_\rho) \, p_{(m,q)}^{(t)}$

**end for**

**for all** $m$ with samples in batch **do**

$\quad p_m^{(t)} \leftarrow \dfrac{\sum_{q \in \mathcal{Q}_m^{(t)}} \sum_{i \in G_{(m,q)}} \left| \hat{A}_{(m,q,i)}^{(t)} \right|}{\sum_{q \in \mathcal{Q}_m^{(t)}} |G_{(m,q)}|}$

$\quad \tilde{p}_m^{(t)} \leftarrow \beta_\rho \, \tilde{p}_m^{(t-1)} + (1 - \beta_\rho) \, p_m^{(t)}$

**end for**

**Construct geometric references:**

**for all** $m$ with samples in batch **do**

$\quad \bar{p}_{\text{ref},m}^{(t)} \leftarrow \left( \prod_{q \in \mathcal{Q}_m^{(t)}} \tilde{p}_{(m,q)}^{(t)} \right)^{1/|\mathcal{Q}_m^{(t)}|}$

**end for**

$\bar{p}_{\text{ref},\mathcal{M}}^{(t)} \leftarrow \left( \prod_{m \in \mathcal{M}} \tilde{p}_m^{(t)} \right)^{1/|\mathcal{M}|}$

**Construct modulation factors:**

**for all** $(m,q)$ in batch **do**

$\quad s_{(m,q)}^{(t)} \leftarrow \bar{p}_{\text{ref},m}^{(t)} / \tilde{p}_{(m,q)}^{(t)}$

$\quad \tilde{s}_{(m,q)}^{(t)} \leftarrow \left( \tilde{s}_{(m,q)}^{(t-1)} \right)^{\beta_s} \left( s_{(m,q)}^{(t)} \right)^{1-\beta_s}$

**end for**

**for all** $m$ with samples in batch **do**

$\quad s_m^{(t)} \leftarrow \bar{p}_{\text{ref},\mathcal{M}}^{(t)} / \tilde{p}_m^{(t)}$

$\quad \tilde{s}_m^{(t)} \leftarrow \left( \tilde{s}_m^{(t-1)} \right)^{\beta_s} \left( s_m^{(t)} \right)^{1-\beta_s}$

**end for**

**Obtain HARPO-modulated advantages:**

**for all** $(m,q)$ in batch **do**

$\quad$ **for all** $i \in G_{(m,q)}$ **do**

$\quad\quad A_{(m,q,i)}^{\text{H}(t)} \leftarrow \tilde{s}_{(m,q)}^{(t)} \, \tilde{s}_m^{(t)} \, \hat{A}_{(m,q,i)}^{(t)}$

$\quad$ **end for**

**end for**

**Optimize policy with HARPO objective:**

Construct PPO-clipped surrogate $\tilde{A}_{(m,q,i):k}^{\text{H}}(\theta)$ from $A_{(m,q,i)}^{\text{H}(t)}$, in the same fashion as Eq. (16)

Update $\theta$ by maximizing $J_{\text{HARPO}}(\theta)$ using $\tilde{A}_{(m,q,i):k}^{\text{H}}(\theta)$ (Eq. (12))

---

# B. Additional Details on the Human Behavioral Atlas Benchmark

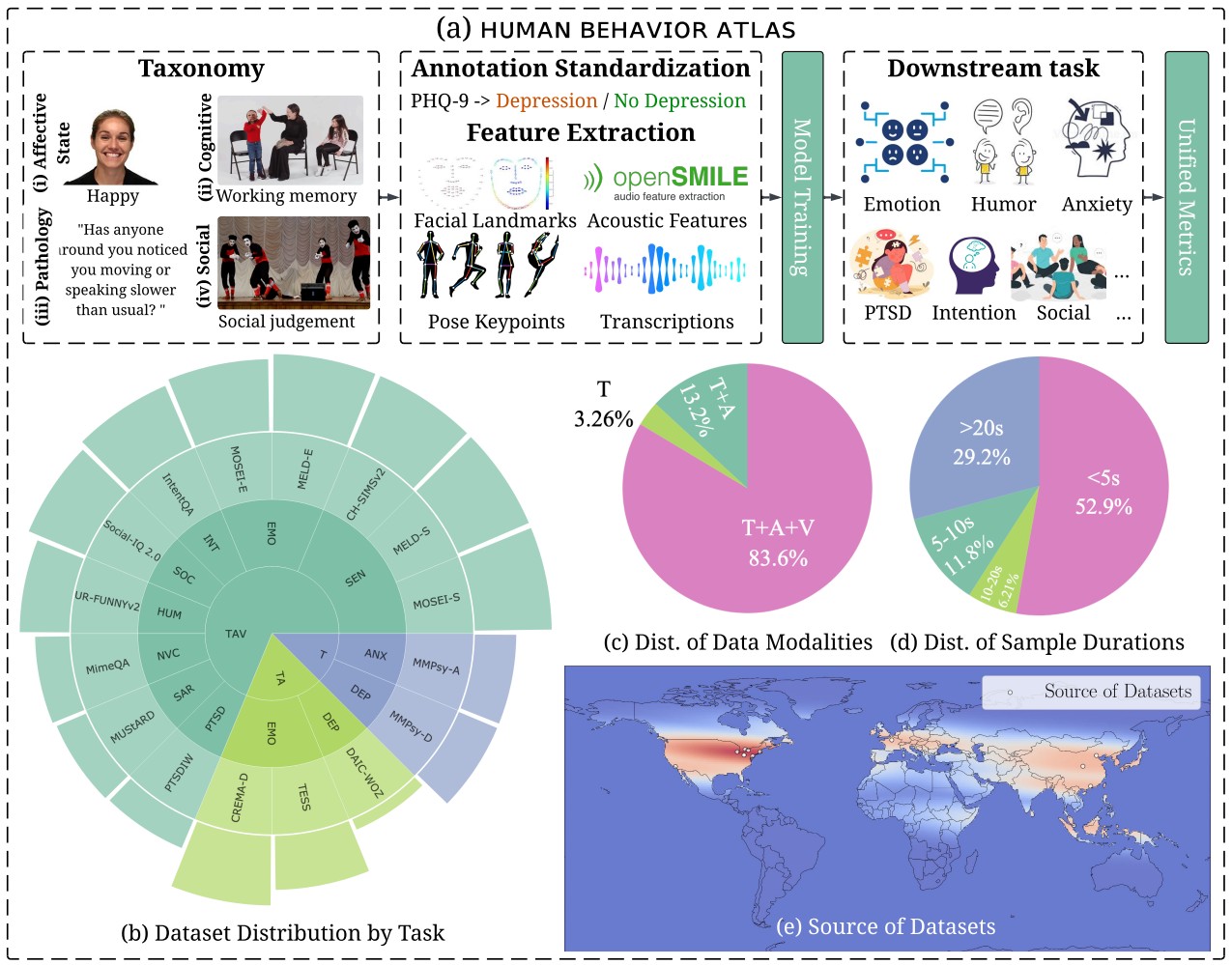

*Figure 7.* Overview of the Human Behavior Atlas (HBA) benchmark (Ong et al., 2026), a large-scale benchmark for unified multimodal social behavior understanding and reasoning. HBA comprises diverse datasets spanning affective, cognitive, pathological, and social behavioral analysis tasks across text, audio, and video modalities. (a) Dataset selection criteria and preprocessing pipeline. (b) Distribution of datasets across 10 behavioral tasks and multimodal input settings. (c) Distribution of data modalities, highlighting the benchmark's strong emphasis on video-centric multimodal understanding. (d) Distribution of sample durations across short- and long-range behavioral understanding scenarios. (e) Geographic distribution of dataset sources across North America, Europe, and Asia.

## B.1. Tasks and Datasets

Human Behavioral Atlas (Ong et al., 2026) comprises approximately 100k samples, with 10 tasks and 13 datasets; we summarize the benchmark below for completeness, with details following the original benchmark description:

The tasks for the benchmark include Sentiment polarity (SEN): classifying attitudes as positive, negative, or neutral; emotion recognition (EMO): identifying emotions (anger, joy, sadness); social reasoning (SOC): understanding socially grounded judgments like empathy or appropriateness; intent recognition (INT): identifying the underlying purpose behind a behavior; and non-verbal communication (NVC): interpreting gestures and facial expressions. They also cover humor detection (HUM), sarcasm detection (SAR), anxiety detection (ANX), depression detection (DEP), and PTSD detection (PTSD).

Each task may be associated with one or more datasets. We summarize the benchmark datasets as follows. CMU-MOSEI (Zadeh et al., 2018) is a large-scale multimodal dataset annotated for sentiment and emotion in real-world opinionated videos. MELD (Poria et al., 2019) provides utterance-level emotion and sentiment annotations from multi-party

dialogues in the TV series *Friends*. UR-FUNNYv2 (Hasan et al., 2019) contains multimodal TED talk clips annotated for humor. MUStARD (Castro et al., 2019) is a multimodal sarcasm detection dataset constructed from television show dialogues. DAIC-WOZ (Valstar et al., 2016) consists of multimodal clinical interviews for depression assessment. CREMA-D (Cao et al., 2014) is an acted emotional speech dataset with categorical emotion labels. CH-SIMSv2 (Yu et al., 2022) comprises multimodal sentiment annotations from simulated human–computer interactions. MMPsy (Zhang et al., 2025b) is a multimodal mental health dataset annotated for anxiety and depression. PTSD in the Wild (Sawadogo et al., 2024) contains real-world videos annotated for post-traumatic stress disorder. TESS (Pichora-Fuller & Dupuis, 2020) is an emotional speech dataset with acted portrayals of discrete emotions. Social-IQ 2 (Wilf et al., 2023) evaluates social intelligence via reasoning over human interactions in video. IntentQA (Li et al., 2023) is a video question-answering dataset focused on intent inference in everyday scenarios. MimeQA (Li et al., 2025) evaluates nonverbal social reasoning using gesture-based video question answering.

We report from the original paper, a Tab. 6 which summarizes the datasets associated task, modalities, sample count and evaluation metric. For completeness, the paper also includes the different dimensions of human behavior that each dataset belongs to. These include, *affective states* (Aff), which capture feelings, emotions and sentiments; *cognitive states* (Cog), which reflect internal mental processes such as reasoning or attention inferred from observable behavior; *pathological states* (Path), which correspond to psychological or psychiatric conditions assessed through verbal or nonverbal indicators; and *social processes* (Soc), which characterize social interaction and communicative behaviors such as humor, intent, and cooperation.

*Table 6.* Datasets and their associated tasks and human behavior dimensions in Human Behavior Atlas. The modalities T, A, V stand for text, audio and vision respectively. The tasks can fall into two categories, CLS = classification (evaluated by direct label matching). TXTR = text-response (evaluated by an LLM judge).

| Dataset | Dimension | Task(s) | Task Type | Modalities | Samples | Eval. Metric |
|---|---|---|---|---|---|---|
| CMU-MOSEI | Aff; Cog | EMO, SEN | CLS | T / A / V | 31,454 | Binary weighted F1 (SEN), Mean weighted acc. (EMO) |
| MELD | Aff; Soc; Cog | EMO, SEN | CLS | T / A / V | 27,412 | Binary weighted F1 (SEN), Mean weighted acc. (EMO) |
| TESS | Aff; Cog | EMO | CLS | T / A / – | 2,800 | Mean weighted accuracy |
| CREMA–D | Aff | EMO | CLS | T / A / – | 7,442 | Mean weighted accuracy |
| CH–SIMSv2 | Aff | SEN | CLS | T / A / V | 4,403 | Binary weighted F1 |
| Social-IQ 2.0 | Soc; Cog | SOC | TXTR | T / A / V | 6,437 | Accuracy (LLM–Judge) |
| IntentQA | Soc; Cog | INT | TXTR | T / A / V | 16,297 | Accuracy (LLM–Judge) |
| MimeQA | Soc | NVC | TXTR | T / A / V | 806 | Accuracy (LLM–Judge) |
| UR–FUNNYv2 | Soc | HUM | CLS | T / A / V | 2,125 | Weighted F1 |
| MUStARD | Soc | SAR | CLS | T / A / V | 690 | Weighted F1 |
| DAIC–WOZ | Path | DEP | CLS | T / A / – | 189 | Weighted F1 |
| MMPsy | Path | DEP, ANX | CLS | T / – / – | 1,275 | Weighted F1 |
| PTSD–in–the–Wild | Path | PTSD | CLS | T / A / V | 634 | Weighted F1 |

## B.2. Evaluation Metrics

We follow the same evaluation metrics utilized in Human Behavior Atlas, which specifically account for the task-specific nuances. The summary of the metrics utilized for each task is found in Tab. 6.

For HUM (Humor Detection), SAR (Sarcasm Detection), DEP (Depression Detection), ANX (Anxiety Detection), and PTSD (PTSD Detection), the weighted F1 score is computed:

$$F1 = \frac{2 \cdot \text{Precision} \cdot \text{Recall}}{\text{Precision} + \text{Recall}},$$

where,

$$\text{Precision} = \frac{TP}{TP + FP}, \qquad \text{Recall} = \frac{TP}{TP + FN}.$$

The weighted F1 is then computed as:

$$\text{Weighted-F1} = \sum_{c \in C} \frac{n_c}{N} \cdot F1_c,$$

where $n_c$ is the number of true instances in class $c$, $N$ is the total number of instances, and $C$ is the set of classes.

SEN (Sentiment Detection) utilizes binary weighted F1, which applies the same formula but only over the positive and negative sentiment classes, where fine-grained sentiment scales (i.e., weakly positive or weakly negative) are mapped into positive or negative classes respectively. This accounts for the differences in sentiment-scale labelling across the SEN datasets.

For EMO (Emotion Recognition), the mean/ average weighted accuracy across all emotion classes (e.g., fear, surprise, joy) is calculated, using the weighted accuracy formula, following Liang et al. (2018):

$$\text{Weighted-Accuracy} = 0.5 \cdot \frac{TP}{P} + 0.5 \cdot \frac{TN}{N},$$

where $TP$ and $TN$ are the number of true positives and true negatives for the target class, and $P$ and $N$ denote the total number of positive and negative samples, respectively.

For free-text response QA tasks such as NVC (Non-Verbal Communication), INT (Intent Recognition), and SOC (Social Reasoning), an LLM judge (GPT-5 nano (OpenAI, 2025)) is used to grade the generated responses. Specifically, task-specific prompts are provided to the LLM judge and the proportion of responses marked as TRUE is recorded as an estimate of accuracy:

$$\text{Accuracy} = \frac{\text{number of TRUE responses}}{\text{number of total responses}}$$

The full grading prompts can be found in the original Human Behavior Atlas paper (Ong et al., 2026).

## C. Experimental Settings.

### C.1. Hyperparameter Settings

We train all methods for up to 5 epochs with early stopping based on validation performance. Validation is performed every 50 training steps. Early stopping is triggered if the validation metric does not improve for 5 consecutive validation runs (i.e., over 250 training steps). This stopping criterion helps reduce the impact of short-horizon fluctuations in validation metrics that arise from the high variance and non-stationarity of on-policy sampling (Henderson et al., 2018). We select the checkpoint achieving the highest average validation weighted F1 score.

From Tab. 1, model results (i.e., Gemma-3-4B (Team et al., 2025), Qwen 2.5-Omni-7B (Xu et al., 2025a), Qwen 2.5-VL-7B (Bai et al., 2025), OmniSapiens-7B RL (Ong et al., 2026), HumanOmniV2-7B (Yang et al., 2025) are taken from the Human Behavior Atlas benchmark paper (Ong et al., 2026)), while we run the evaluation of Qwen 3-VL-8B Instruct (Xu et al., 2025b) using its implementation on Huggingface[4].

On the other hand, we implement the reinforcement learning training algorithms in Tab. 1 (i.e., RLOO (Ahmadian et al., 2024), RE++ (Hu, 2025), GPG (Chu et al., 2025), GRPO (Shao et al., 2024)) using the VERL package [5]. To enable fair comparison with HARPO, all reinforcement learning methods are run on the same Human Behavior Atlas benchmark, with the same base model, Qwen 2.5-Omni-7B (Xu et al., 2025a), and exactly the same reward design in Sec. 3.4. For the RL baselines, we follow standard practice by reusing the hyperparameter configurations reported in the original papers, thereby preserving the authors' intended optimization settings. For HARPO, we retain the learning rate used by GRPO ($1 \times 10^{-6}$), since HARPO retains the update structure as GRPO, and we set $\beta_\rho$, $\beta_s$ in the inertial control Eq. (11) as 0.95. We also utilize the AdamW optimizer and omit explicit KL regularization. This is motivated by HARPO's advantage modulation, which already acts to regularize policy updates, and is aligned with prior literature that strict KL constraints are not always necessary in on-policy training (Yu et al., 2025). For all reinforcement learning methods, we fix the number of rollouts to 5, utilize the same effective batch size of 256 with PPO-mini-batch size of 128, and set the prompt and response lengths to 4096 and 2048 respectively. All methods are run on 4 Nvidia H200s as well as 4 Nvidia RTX PRO 6000 Blackwell GPUs.

### C.2. Full Details on Reward Design

---

[4]https://huggingface.co

[5]https://github.com/volcengine/verl

From Sec. 3.4, we utilize a final reward per sample that combines task accuracy $r_{task}$, format correctness $r_{fmt}$, length penalty $r_{len}$. We summarize this below, with format weight $w_{\mathrm{fmt}} = 0.2$ and length scale $\lambda_{\mathrm{len}} = 0.75$, where $r_{task}$ can be $r_{cls}$ or $r_{qa}$, depending on if the sample belongs to a classification or QA task respectively:

$$r = (1 - w_{\mathrm{fmt}})\, r_{\mathrm{task}} + w_{\mathrm{fmt}}\, r_{\mathrm{fmt}} + \lambda_{\mathrm{len}}\, r_{\mathrm{len}},$$

We provide additional details on $r_{task}, r_{fmt}, r_{len}$. For $r_{task}$, it can either be classification reward $r_{cls}$, or question-answering reward $r_{qa}$, depending on whether the sample involves a classification response or a free-text answer respectively. $r_{cls}$ is computed using a binary score for whether the predicted label exactly matches the ground truth label. $r_{qa}$ leverages a cosine similarity reward that compares the embedding of the predicted free text sequence and the ground truth, leveraging MiniLM-L6-v2[6] as the embedding model. Because cosine similarity typically falls between $[-1, 1]$, we ensure that the values are transformed into the range of $[0, 1]$, such that it is compatible with reward assignment. We sumamarise this below, where $\hat{y}$ is the predicted response (which contains just the answer without reasoning trace) and $y$ is the ground truth:

$$r_{\mathrm{task}}(\hat{y}, y) = \begin{cases} r_{\mathrm{cls}}(\hat{y}, y), & \text{if classification,} \\ r_{\mathrm{qa}}(\hat{y}, y), & \text{if free-text QA.} \end{cases}$$

$$r_{\mathrm{cls}}(\hat{y}, y) = \begin{cases} 1, & \hat{y} = y, \\ 0, & \text{otherwise.} \end{cases}$$

$$r_{\mathrm{qa}}(\hat{y}, y) = \frac{\cos(e(\hat{y}), e(y)) + 1}{2}, \qquad \cos(u, v) = \frac{u^\top v}{\|u\|_2\, \|v\|_2}, \qquad u = e(\hat{y}),\ v = e(y).$$

For the formatting reward $r_{fmt}$, we assign a binary reward based on whether the generated response strictly follows the output format specified in the prompt. In particular, we provide the model the following system prompt:

```
{content}
You FIRST think about the reasoning process as an internal monologue and then
    provide the final answer. The reasoning process MUST BE enclosed within <
    think> </think> tags. The final answer MUST BE put in \boxed{}.
```

A response receives $r_{fmt} = 1$ if and only if it contains `<think>` and `</think>` tags, followed by a final prediction enclosed in `\boxed`, in this order; otherwise, $r_{\mathrm{fmt}} = 0$. This check is applied to the full response and enforces adherence to the prescribed output schema.

Finally, we provide a overlong length penalty, $r_{len}$ which follows Zhang & Zuo (2025) to prevent excessive length and verbosity of responses. Accordingly, let $\ell$ be response length, we use a max $L_{\mathrm{max}}$ of 812 tokens and buffer length $B$ of 128 tokens. The penalty is:

$$r_{\mathrm{len}}(\ell) = \begin{cases} 0, & \ell \leq L_{\mathrm{max}} - B, \\ \frac{(L_{\mathrm{max}} - B) - \ell}{B}, & L_{\mathrm{max}} - B < \ell \leq L_{\mathrm{max}}, \\ -1, & \ell > L_{\mathrm{max}}. \end{cases}$$

## D. Reasoning Evaluation Setup

We provide the full details of the setup for evaluating the reasoning trace interpretability in the following. Metrics are computed over the union of held-out zero-shot tasks (AUT, SER, IDR, SMSA, SIR), with all models evaluated under identical prompt templates, data loading, and multimodal preprocessing. Generation is capped at $T_{\mathrm{max}} = 512$ new tokens per sample.

**Consistency.** For each sample $x$, $N = 5$ independent reasoning traces $\{\hat{y}_i(x)\}_{i=1}^N$ are drawn with sampling parameters set to each model's recommended decoding configuration, where $T$ denotes the softmax temperature, $p$ the nucleus (top-$p$)

---

[6]https://huggingface.co/sentence-transformers/all-MiniLM-L6-v2

cumulative probability threshold, and $k$ the top-$k$ truncation cutoff. We use $(T, p, k) = (0.6, 0.95, 20)$ for Qwen2.5-Omni-7B, HumanOmniV2, OmniSapiens-7B-RL, and OMNISAPIENS-7B 2.0, and $(T, p, k) = (1.0, 0.95, 64)$ for Gemma 4-E4B. Consistency is the fraction of instances on which $N$ independent reasoning traces produce the same answer for a given sample $x$, averaged across all samples:

$$\text{Consistency} \;=\; \frac{1}{|\mathcal{D}|} \sum_{x \in \mathcal{D}} \mathbb{K}\big[\,|\{\hat{y}_1(x), \ldots, \hat{y}_N(x)\}| = 1\,\big], \tag{13}$$

**Average tokens.** Average tokens are computed on the same $N$ traces by extracting the reasoning content between the model's thinking delimiters and tokenising with each model's own tokenizer, so that reported lengths reflect the actual inference cost incurred per model.

**Paraphrase.** A single reasoning trace is drawn under the same stochastic decoding configuration (as per consistency evaluation setup) on a semantically equivalent paraphrase $\tilde{x}$ of the original prompt, yielding $\hat{y}_{\text{para}}(x)$. Paraphrased prompts are pre-generated offline (one paraphrase per test instance), and held fixed across all models so that robustness is measured under the same input perturbation for every method; only the natural-language instruction is paraphrased, while task labels, output format, and multimodal inputs are preserved verbatim. Paraphrase consistency is the agreement between the original and paraphrased predictions, where $\hat{y}(x)$ denotes the first stochastic sample on the original prompt under the same decoding parameters, and we average this score over all samples. We provide the paraphrased prompts in the code release: https://github.com/MIT-MI/human_behavior_atlas.

$$\text{Paraphrase} \;=\; \frac{1}{|\mathcal{D}|} \sum_{x \in \mathcal{D}} \mathbb{K}\big[\,\hat{y}(x) = \hat{y}_{\text{para}}(x)\,\big], \tag{14}$$

**Human evaluation.** Three annotators with backgrounds in interpretability, human–computer interaction, and healthcare, domains directly relevant to the critical applications of behavioral models, conducted pairwise comparisons between OMNISAPIENS-7B 2.0 and each baseline. For each instance, annotators are shown two reasoning traces (A and B) extracted from the `<think>...</think>` block of each model, with the task question provided as optional context, and select A, B, or Tie independently along three criteria mapped to XAI desiderata (Alvarez Melis & Jaakkola, 2018): specificity (completeness), coherence (understandability), and concision (compactness). For each baseline $b$ and criterion $c$, let $w_b^c, \ell_b^c, t_b^c$ denote the counts of OMNISAPIENS-7B 2.0-preferred, baseline-preferred, and tied judgments with $n_b^c = w_b^c + \ell_b^c + t_b^c$; we report the rates for Tie+Win $= (w_b^c + t_b^c)/n_b^c$, Tie $= t_b^c/n_b^c$, and Win$^\dagger = w_b^c/(w_b^c + \ell_b^c)$, where Win$^\dagger$ isolates the win rate excluding ties. Aggregate rates in Tab. 4 average across baselines. The annotation instructions shown in the box below. To ensure fair and robust evaluation, the annotators were not part of the authorship of this paper, and the OMNISAPIENS-7B 2.0 and baselines are masked with letters A and B.

---

**Annotation Instructions**

You will be shown pairs of reasoning explanations (A and B) for the same prediction. The reasoning is between the `<think></think>` tags. Your task is to compare them and select which explanation is better along each criterion. For each row, choose **A** if Explanation A is better, **B** if Explanation B is better, or **Tie** if they are similar. Place A, B, or Tie under the Specificity, Coherence, and Concision columns respectively.

**Evaluation Criteria.**

- **Specificity:** Which explanation is more concrete and avoids vague or generic statements?
- **Coherence:** Which explanation is more logically structured and easier to follow?
- **Concision:** Which explanation conveys its key points more efficiently without unnecessary verbosity?

**Notes.** Focus only on the quality of the reasoning explanation. You may consult the task question for optional context. Evaluate based on the explanations themselves. Work quickly and rely on your first judgment.

---

# E. Additional Formulas

## E.1. Full GRPO Formulation

For completeness, we provide the full formulation of GRPO in Sec. 3.2, including the surrogate objective and importance-sampling formulation.

Accordingly, for task $m$ and sample $q$, GRPO samples a rollout group $G_{(m,q)}$ of responses $\{o_{(m,q,i)}\}$, where $i \in G_{(m,q)}$ indexes individual rollouts (i.e., a sampled response) with rewards $r_{(m,q,i)}$, computing the group-normalized advantage:

$$\hat{A}_{(m,q,i)} = \frac{r_{(m,q,i)} - \hat{\mu}_{G_{(m,q)}}}{\hat{\sigma}_{G_{(m,q)}} + \varepsilon}, \tag{15}$$

where $\hat{\mu}_{G_{(m,q)}}$ and $\hat{\sigma}_{G_{(m,q)}}$ are the empirical mean and standard deviation of $\{r_{(m,q,i)}\}_{i=1}^{|G_{(m,q)}|}$. GRPO then optimizes $\pi_\theta(a \mid s)$ by performing a PPO-style trust-region update. At token position $k$ of response $o_{(m,q,i)}$, $\varphi_{(m,q,i):k}(\theta)$ denotes the importance sampling ratio between $\pi_\theta$ and the old policy $\pi_{\theta_{\text{old}}}$, $\tilde{A}_{(m,q,i):k}(\theta)$ denotes the PPO-clipped surrogate using $\hat{A}_{(m,q,i)}$, and $J_{\text{GRPO}}(\theta)$ averages this surrogate over tokens and rollout samples with an optional KL penalty to a reference policy $\pi_{\text{ref}}$ (with weight $\beta$). We summarize these with a compact objective:

$$\varphi_{(m,q,i):k}(\theta) = \frac{\pi_\theta\big(o_{(m,q,i):k} \mid q,\, o_{(m,q,i):<k}\big)}{\pi_{\theta_{\text{old}}}\big(o_{(m,q,i):k} \mid q,\, o_{(m,q,i):<k}\big)}$$

$$\tilde{A}_{(m,q,i):k}(\theta) = \min\Big(\varphi_{(m,q,i):k}(\theta)\,\hat{A}_{(m,q,i)},$$

$$\text{clip}\big(\varphi_{(m,q,i):k}(\theta),\, 1 - \epsilon,\, 1 + \epsilon\big)\hat{A}_{(m,q,i)}\Big) \tag{16}$$

$$J_{\text{GRPO}}(\theta) = \mathbb{E}_{q \sim \mathcal{D}_m} \mathbb{E}_{\{o_{(m,q,i)}\} \sim \pi_{\theta_{\text{old}}}} \left[ \frac{1}{|G_{(m,q)}|} \sum_{i=1}^{|G_{(m,q)}|} \right.$$

$$\left. \frac{1}{n_{o_{(m,q,i)}}} \sum_{k=1}^{n_{o_{(m,q,i)}}} \tilde{A}_{(m,q,i):k}(\theta) \right] - \beta\, \mathbb{E}[D_{\text{KL}}(\pi_\theta \parallel \pi_{\text{ref}})].$$

### E.2. Additional HARPO Details

In Sec. 3.3, we explained the structured modulation utilized by HARPO, which constructs modulation factors by comparing contribution signals to a geometric mean reference. We provide further details on why this construction yields modulation factors whose geometric mean equals 1, ensuring that multiplicative upscaling from certain modulation factors is exactly compensated by downscaling from others.

Fix an iteration $t$ and a task $m$, and let $\mathcal{Q}_m^{(t)}$ denote the set of samples associated with task $m$. The sample-level geometric reference is defined as:

$$\bar{p}_{\text{ref},m}^{(t)} = \left( \prod_{q \in \mathcal{Q}_m^{(t)}} p_{(m,q)}^{(t)} \right)^{\frac{1}{|\mathcal{Q}_m^{(t)}|}}.$$

Using this reference, the sample-level modulation factor for each sample $q$ is constructed as:

$$s_{(m,q)}^{(t)} = \frac{\bar{p}_{\text{ref},m}^{(t)}}{p_{(m,q)}^{(t)}}.$$

Taking the product over all samples in $\mathcal{Q}_m^{(t)}$ yields:

$$\prod_{q \in \mathcal{Q}_m^{(t)}} s_{(m,q)}^{(t)} = \prod_{q \in \mathcal{Q}_m^{(t)}} \frac{\bar{p}_{\text{ref},m}^{(t)}}{p_{(m,q)}^{(t)}} = \frac{(\bar{p}_{\text{ref},m}^{(t)})^{|\mathcal{Q}_m^{(t)}|}}{\prod_{q \in \mathcal{Q}_m^{(t)}} p_{(m,q)}^{(t)}}.$$

By definition of the geometric mean:

$$(\bar{p}_{\text{ref},m}^{(t)})^{|\mathcal{Q}_m^{(t)}|} = \prod_{q \in \mathcal{Q}_m^{(t)}} p_{(m,q)}^{(t)},$$

and therefore:
$$\prod_{q \in \mathcal{Q}_m^{(t)}} s_{(m,q)}^{(t)} = 1.$$

This shows that the sample-level modulation factors possess a geometric mean of 1.

An identical argument applies at the task-level. Let $\mathcal{M}$ denote the set of tasks and define the task-level geometric reference as
$$\bar{p}_{\text{ref},\mathcal{M}}^{(t)} = \left( \prod_{m \in \mathcal{M}} p_m^{(t)} \right)^{\frac{1}{|\mathcal{M}|}}.$$

The task-level modulation factors are given by
$$s_m^{(t)} = \frac{\bar{p}_{\text{ref},\mathcal{M}}^{(t)}}{p_m^{(t)}}.$$

Taking the product over all tasks yields
$$\prod_{m \in \mathcal{M}} s_m^{(t)} = 1.$$

Therefore, the modulation factors are possess a geometric mean of 1 at both the sample and task-levels, ensuring that multiplicative upscaling from some modulation factors is exactly compensated by downscaling from others. Hence, the factors cannot simulateneously enlarge or shrink all advantages at the sample or task-level, mitigating unintended influence on the effective global step size.

# F. Additional Results

## F.1. Full Ablation Results

We provide the per task and per dataset breakdown of the results from the ablation in the following. Tab. 7 represents the task-level performance of the ablations, whereas Tab. 8 represents the per-dataset performance breakdown of all ablations.

*Table 7.* Per-task performance (%) for HARPO ablations across behavioral tasks. Each value is the arithmetic mean over datasets associated with the task. Avg. Rank is computed across tasks using per-task ranks (higher is better; ties use average rank), and then averaged over tasks (lower is better). Rows with missing task values are omitted from ranking.

| Variant | EMO | HUM | INT | PTSD | ANX | DEP | SEN | SAR | SOC | NVC | Avg. Rank ↓ |
|---|---|---|---|---|---|---|---|---|---|---|---|
| **HARPO** | 76.55 | **69.85** | 50.52 | **98.39** | **91.98** | **78.87** | **77.61** | 70.64 | 25.40 | 14.54 | **1.90** |
| w/o structured modulation | **78.44** | 66.61 | 50.38 | 98.39 | 90.68 | 77.01 | 77.48 | **72.68** | 29.08 | **15.12** | 2.00 |
| w/o inertial control | 76.42 | 63.05 | **53.28** | 98.39 | 90.68 | 76.48 | 71.92 | 62.68 | **30.12** | 13.95 | 2.70 |
| w/o sample-level modulation | 77.27 | 67.82 | 50.94 | 98.39 | 90.68 | 71.04 | 76.37 | 68.19 | 29.43 | 12.21 | 2.60 |

*Table 8.* Ablation results for HARPO components, grouped by behavioral tasks and datasets. Following the unified metrics proposed in the Human Behavior Atlas Benchmark (Ong et al., 2026), we use binary weighted F1 for SEN; mean per-class weighted accuracy for EMO; weighted F1 for HUM, SAR, ANX, DEP, PTSD; and LLM-Judge accuracy for SOC, INT, NVC.

| Model / Ablations | EMO | | | | HUM | INT | PTSD | ANX | DEP | | SEN | | | SAR | SOC | NVC |
|---|---|---|---|---|---|---|---|---|---|---|---|---|---|---|---|---|
| | CREMA-D | MELD (E) | MOSEI (E) | TESS | UR-FUNNY | IntentQA | PTSD_WILD | MMPSY (A) | MMPSY (D) | DAIC-WOZ | MELD (S) | CH-SIMSv2 | MOSEI (S) | MUStARD | Social-IQ 2.0 | MimeQA |
| HARPO | 85.80 | 69.14 | 55.45 | 95.83 | 69.85 | 50.52 | 98.39 | 91.98 | 84.53 | 73.20 | 75.49 | 88.71 | 68.64 | 70.64 | 25.40 | 14.54 |
| w/o structured modulation | 84.48 | 71.70 | 60.36 | 97.22 | 66.61 | 50.38 | 98.39 | 90.68 | 83.20 | 70.82 | 70.43 | 88.37 | 73.65 | 72.68 | 29.08 | 15.12 |
| w/o inertial control | 86.49 | 67.86 | 56.61 | 94.72 | 63.05 | 53.28 | 98.39 | 90.68 | 82.04 | 70.92 | 73.80 | 84.65 | 57.32 | 62.68 | 30.12 | 13.95 |
| w/o sample-level modulation | 84.71 | 68.92 | 57.38 | 98.06 | 67.82 | 50.94 | 98.39 | 90.68 | 73.44 | 68.64 | 71.61 | 87.76 | 69.74 | 68.19 | 29.43 | 12.21 |

## F.2. Full Dataset Results

For brevity, Tab. 1 shows the performance at the task-level. We additionally provide the dataset-level breakdown of these results in the following Tab. 9. We also provided the full dataset breakdown for HARPO and its ablations in Tab. 8.

## F.3. Additional Training Plots

We provide additional training plots to empirically illustrate the training dynamics in our experiments.

We include a Fig. 9 that depicts the advantage distributions of the different behavioral tasks in the Human Behavior Atlas benchmark (Ong et al., 2026). In particular, the right column of Fig. 9 depicts the group-normalized advantages of the tasks SEN, NVC, INT under a GRPO run. The different advantage distributions observed highlights how the behavioral tasks can systematically induce different advantages, as mentioned in Sec. 3.2.

In the left column of this Fig. 9, we compare the advantage distributions with and without sample-level modulation for other tasks (SOC, HUM, NVC). Accordingly, we observe that with sample-level modulation in the HARPO method, the advantage distributions tend to become narrower than without. This coincides with more consistent performance across tasks in Tab. 5 with HARPO achieving an average task rank of 1.90 compared to without sample-level modulation at 2.60.

We include a Fig. 8 to illustrate the task-level modulation factors over time between the ablation that utilizes $s^{(t)} = 1/p^{(t)}$ and HARPO. To this end, we observe that in the left column of this figure, the geometric mean of the modulation factors are consistently above 2.5. This further illustrates how the modulation factors can induce unintended scaling of the global step size, without geometric centering.

In the same Fig. 8, in the right column, we also observe how the contribution signals $p^{(t)}$ varies with training for different tasks, SAR, INT, EMO, SOC. Accordingly, we observe that for certain tasks, the task-level contribution signals $p_m^{(t)}$ can differ by orders of magnitude. For example, the $p_m^{(t)}$ for INT may exceed EMO by a factor of up to 7. Since these signals are used to construct the modulation factors, the empirical illustrations provide additional context for the considerable

*Table 9.* Full results grouped by behavioral tasks (headers) and their relevant datasets (sub-headers). Following the unified metrics proposed in the Human Behavior Atlas Benchmark (Ong et al., 2026), we use binary weighted F1 for SEN; mean per-class weighted accuracy for EMO; weighted F1 for HUM, SAR, ANX, DEP, PTSD; and LLM-Judge accuracy for SOC, INT, NVC.

| Model / Algorithm | EMO | | | | HUM | INT | PTSD | ANX | DEP | | SEN | | | SAR | SOC | NVC |
|---|---|---|---|---|---|---|---|---|---|---|---|---|---|---|---|---|
| | CREMA-D | MELD (E) | MOSEI (E) | TESS | UR-FUNNY | IntentQA | PTSD_WILD | MMPSY (A) | MMPSY (D) | DAIC–WOZ | MELD (S) | CH-SIMSv2 | MOSEI (S) | MUStARD | Social-IQ 2.0 | MimeQA |
| **Models** | | | | | | | | | | | | | | | | |
| Gemma-4-E4B (8B) | 50.21 | 66.02 | 58.63 | 49.07 | 52.63 | 40.25 | 86.00 | 33.36 | 16.05 | 29.05 | 68.89 | 76.53 | 63.24 | 73.54 | 20.40 | 4.65 |
| Gemma-3-4B | 49.50 | 64.20 | 56.50 | 49.90 | 59.70 | 22.70 | 49.90 | 60.10 | 78.80 | 13.70 | 78.50 | 81.30 | 61.70 | 52.90 | 19.10 | 2.30 |
| Qwen 2.5-Omni-7B | 52.10 | 66.10 | 58.00 | 56.80 | 54.30 | 25.40 | 76.00 | 79.30 | 79.10 | 63.60 | 70.00 | 71.40 | 60.20 | 65.60 | 25.40 | 6.90 |
| Qwen-2.5-VL-7B | 50.10 | 57.10 | 59.20 | 49.90 | 58.30 | 24.90 | 75.50 | 63.10 | 65.30 | 62.30 | 67.40 | 52.40 | 31.70 | 51.10 | 23.10 | 9.80 |
| Qwen 3-VL-8B-Instruct | 50.20 | 70.87 | 59.44 | 50.14 | 66.76 | 38.00 | 92.70 | 42.29 | 36.38 | 66.86 | 68.97 | 77.19 | 62.93 | 63.67 | 24.94 | 13.95 |
| OMNISAPIENS-7B RL | 50.10 | 69.90 | 58.10 | 51.00 | 63.90 | 48.60 | 96.80 | 91.90 | 81.40 | 72.90 | 57.10 | 39.30 | 22.40 | 64.70 | 30.40 | 13.30 |
| HumanOmniV2-7B | 56.00 | 63.30 | 55.80 | 63.70 | 63.80 | 26.30 | 82.40 | 52.70 | 67.20 | 63.60 | 76.80 | 82.50 | 63.30 | 39.50 | 28.20 | 9.30 |
| **OMNISAPIENS-7B 2.0 (ours)** | 85.80 | 69.14 | 55.45 | 95.83 | 69.85 | 50.52 | 98.39 | 91.98 | 84.53 | 73.20 | 75.49 | 88.71 | 68.64 | 70.64 | 25.40 | 14.54 |
| **Training Algorithms** | | | | | | | | | | | | | | | | |
| RLOO | 85.77 | 67.74 | 50.05 | 98.75 | 67.86 | 51.73 | 98.39 | 90.68 | 84.26 | 70.88 | 67.72 | 89.64 | 73.21 | 62.58 | 29.54 | 16.28 |
| RE++ | 82.56 | 66.52 | 59.05 | 95.56 | 60.26 | 5.01 | 98.39 | 93.11 | 79.09 | 68.64 | 66.67 | 87.86 | 15.03 | 50.21 | 12.64 | 4.07 |
| GPG | 82.62 | 69.36 | 60.00 | 98.75 | 69.28 | 54.21 | 98.39 | 90.36 | 85.88 | 70.92 | 75.10 | 87.77 | 64.43 | 45.96 | 27.93 | 12.79 |
| GRPO | 82.88 | 69.85 | 57.22 | 95.83 | 27.56 | 49.90 | 98.39 | 90.42 | 84.35 | 70.92 | 76.28 | 84.57 | 71.68 | 53.58 | 23.30 | 11.00 |
| EMA-GRPO | 76.21 | 65.37 | 56.85 | 95.00 | 63.50 | 52.62 | 98.39 | 90.68 | 75.84 | 70.88 | 65.94 | 76.94 | 61.96 | 77.75 | 30.58 | 15.70 |
| **HARPO (ours)** | 85.80 | 69.14 | 55.45 | 95.83 | 69.85 | 50.52 | 98.39 | 91.98 | 84.53 | 73.20 | 75.49 | 88.71 | 68.64 | 70.64 | 25.40 | 14.54 |

variation in modulation factors. This motivates the use of a geometric reference as a practical tool to temper excessively large differences in modulation factors.

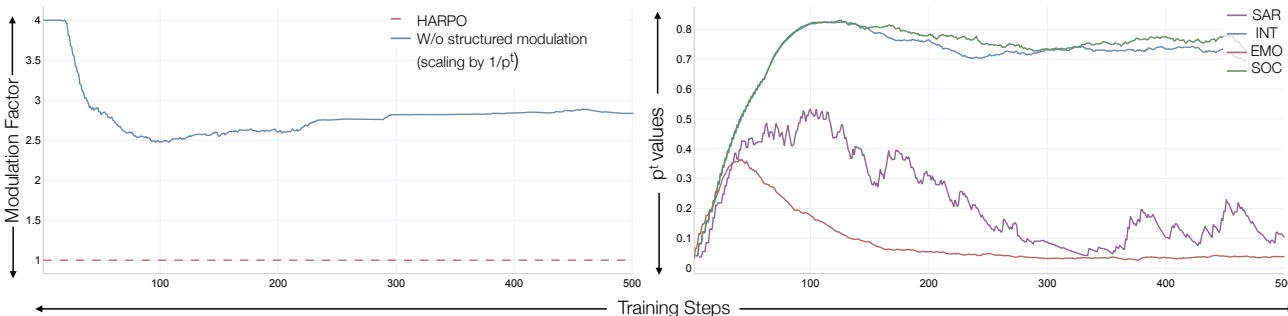

*Figure 8.* Left: Comparison of geometric mean of the task-level modulation factors $s_m^{(t)}$ between HARPO and the w/o structured modulation ablation that utilizes $s^{(t)} = 1/p^{(t)}$. We observe that the modulation geometric mean of the modulation factors are above 2.5 throughtout training. Right: Values of $p^{(t)}$ over training for different tasks SAR, INT, EMO, SOC. We observe that between specific tasks (i.e., INT vs EMO), the task-level contribution signals $p_m^{(t)}$ can vary by considerable orders of magnitude throughout training.

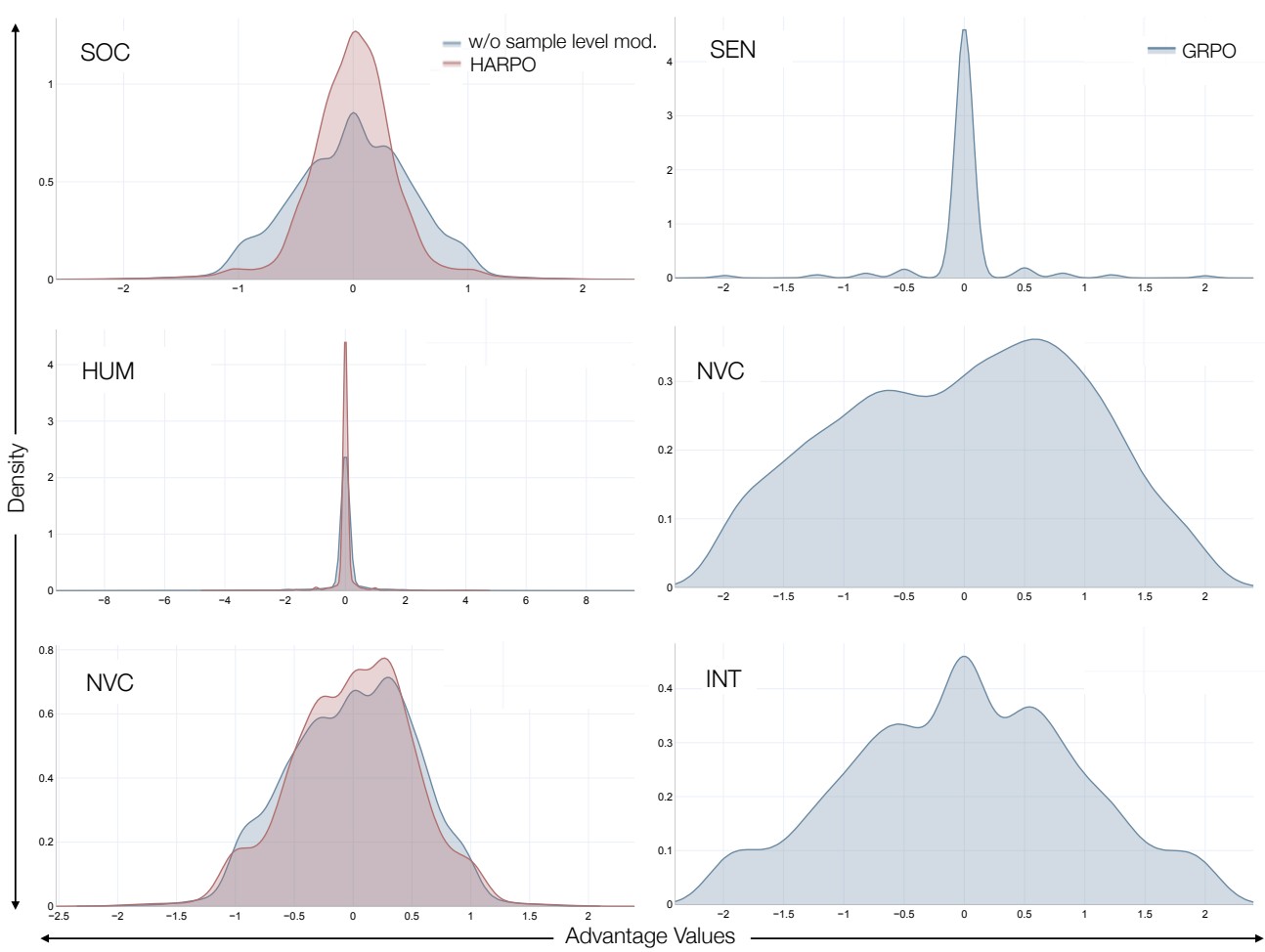

*Figure 9.* Left: Advantage distributions for HARPO (red) and the ablation (blue) which does not involve sample-level modulation, across the different tasks SOC, HUM, NVC. We observe that HARPO, with sample-level modulation induces a narrower advantage distribution. Right: More plots to depict the group-normalized advantage distributions of different tasks, SEN, NVC, INT. We observe that the different behavioral tasks provide different advantage distributions.

