# OpenReview forum: "OmniSapiens: A Foundation Model for Social Behavior Processing via Heterogeneity-Aware Relative Policy Optimization"
_ICML.cc/2026/Conference — ICML 2026 regular_

### Official Review · Reviewer_SL1F · 2026-03-13

**Soundness:** 3
**Presentation:** 3
**Significance:** 4
**Originality:** 3
**Overall Recommendation:** 3
**Confidence:** 5

**Summary:**

The authors tackle the challenge of developing a unified foundation model capable of processing diverse human social behaviors (emotion, intent, mental states, etc.). They point out a major bottleneck in applying recent reasoning-based RL methods (like GRPO) to this multitask domain: the extreme heterogeneity of behavioral data. Because the tasks span different modalities and prediction targets (e.g., classification vs. free-text QA), their advantage distributions vary wildly. Consequently, when trained jointly, tasks or samples with systematically larger advantage magnitudes disproportionately dominate the policy gradient updates, leading to uneven learning and performance collapse on weaker tasks.

To address this, the paper introduces Heterogeneity-Aware Relative Policy Optimization (HARPO). The core mechanism is an on-policy dynamic advantage modulation applied before the policy update. HARPO estimates "contribution signals" for both individual samples and entire tasks based on their advantage magnitudes. It then scales these advantages using geometrically centered and inertially smoothed modulation factors. This elegantly ensures that no single task or sample overwhelms the optimization step, forcing the model to learn in a balanced way across the heterogeneous data distribution.

Empirically, the authors applied HARPO to fine-tune a Qwen-2.5-Omni-7B base model on the Human Behavior Atlas benchmark, yielding OmniSapiens-7B 2.0. Comprehensive experiments demonstrate that OmniSapiens-7B 2.0 consistently outperforms both general-purpose LLMs/VLMs and existing specialized behavioral models across 10 diverse tasks. Furthermore, the method exhibits strong zero-shot transfer capabilities on held-out datasets (AUT and SER). Qualitatively, the balanced optimization from HARPO prevents the suppression of useful reasoning trajectories, inducing richer, more explicit, and evidence-grounded reasoning traces, particularly for complex pragmatic tasks like humor and sarcasm.

**Compliance With Llm Reviewing Policy:**

Affirmed.

**Ethical Review Flag:**

Flag this paper for an ethics review.

**Key Questions For Authors:**

1. The catastrophic failure of standard GRPO on specific tasks (e.g., SAR) strongly suggests a cold-start issue due to a lack of domain knowledge in the base model, rather than an algorithmic failing. Did you conduct a proper SFT phase before RL? If so, please provide the details. If not, can you provide an ablation showing the performance of both GRPO and HARPO starting from an SFT-warmed checkpoint? I am concerned that HARPO's relative gains might disappear with proper SFT.
2. HARPO's formulation is a generic heterogeneous multi-task optimizer. Why restrict its framing and evaluation strictly to social behaviors when it contains zero social priors? Have you evaluated HARPO on standard, non-social multi-task reasoning benchmarks (e.g., combining Math, Code, and Logic)?
3. How exactly were datasets requiring audio (e.g., CREMA-D, TESS) evaluated on pure Vision-Language baselines (like Qwen 2.5-VL and Qwen 3-VL) that lack an audio encoder? Were ASR transcripts provided, or was the audio modality ignored?
4. How does HARPO compare against other recent LLM-adapted multi-task gradient balancing techniques, such as EMA-GRPO (*OneThinker*), *CTWA* for cross-objective interference, *Multi-Objective GRPO*, or *Dynamic Reward Weighting*? An empirical or theoretical comparison against these concurrent methods is necessary to prove HARPO's superiority as a multi-task optimizer.

**Limitations:**

yes

**Strengths And Weaknesses:**

The paper’s core algorithmic contribution is undeniably elegant. Using geometrically centered and inertially smoothed modulation factors to dynamically balance advantage values in multi-task GRPO is a smart, mathematically sound approach. It directly tackles the gradient dominance issue when training on highly heterogeneous data. The fact that OmniSapiens-7B 2.0 achieves strong results across various public datasets proves the effectiveness of this optimization strategy.

However, there is a severe mismatch between the mathematical formulation and the paper's framing. HARPO (Equations 6-12) is fundamentally a completely generic gradient-balancing optimizer for multi-task RL. It contains absolutely zero social inductive biases, psychological priors, or domain-specific architectural designs. Packaging this purely general RL technique exclusively as a "Foundation Model for Social Behavior" is a significant overclaim. The mechanism has no unique specificity to social tasks; it should theoretically apply to any heterogeneous multi-task setup. By artificially restricting the scope to social datasets, the authors ironically obscure the true value of their method.

Experimentally, the complete absence of a Supervised Fine-Tuning (SFT) phase raises a critical red flag that threatens the paper's main conclusions. The authors note that standard GRPO completely collapsed on tasks like SAR (Sarcasm). However, this catastrophic failure is highly likely a "cold-start" problem rather than a fundamental flaw of standard GRPO. The base model likely lacks exposure to these specific, complex social reasoning tasks during pre-training. Without a proper SFT warm-up to instill basic task comprehension and format adherence, any RL algorithm will struggle. It is entirely possible that if both GRPO and HARPO started from a properly supervised SFT checkpoint, the baselines would not collapse, and HARPO's claimed superiority might vanish or diminish significantly.

Furthermore, the experimental baselines suffer from glaring modality unfairness. The authors compare their model against audio-blind Vision-Language Models (like Qwen 2.5-VL and Qwen 3-VL) on datasets that strictly require audio (e.g., CREMA-D, TESS). Without any explanation of how these missing modalities were handled (e.g., using ASR transcripts or just dropping the audio entirely), these baseline comparisons are fundamentally flawed.

Additionally, the comparative baselines for the multi-task RL optimization are insufficient. The authors miss several highly relevant, recent works specifically tailored to resolving multi-task/multi-objective conflicts in LLM alignment and GRPO. For instance, methods like EMA-GRPO introduced in *OneThinker* (Dec 2025), *CTWA (Covariance-Targeted Weight Adaptation)* for cross-objective interference (Feb 2026), the *Multi-Objective GRPO* framework (Mar 2025), and *Dynamic Reward Weighting* for RL alignment (Sep 2025) all tackle the exact same problem of task imbalance. Failing to compare HARPO against these specialized multi-task conflict resolution techniques significantly weakens the empirical validation of the proposed algorithm. Finally, regarding the social evaluation itself, it would be a nice addition to briefly include or discuss recent causal and social video datasets like *Read the Room* and *CausalChaos!* to further strengthen the video reasoning assessment.

The authors have not adequately addressed the methodological limitations of their experimental design. The most critical limitation is the omission of an SFT (Supervised Fine-Tuning) baseline discussion. By training a base model directly with GRPO on unseen, complex tasks, the paper risks solving an artificially created "cold-start" problem, masking whether HARPO is genuinely superior to standard GRPO when models are properly initialized. Secondly, the paper fails to acknowledge the limitation of comparing their multi-modal model against audio-blind VLMs without ensuring modality parity, compromising the integrity of those baseline results. Finally, claiming the model is specifically a "Foundation Model for Social Behavior" limits the scope of a fundamentally general mathematical optimizer, which is a conceptual limitation of the paper's framing.

I am very open to raising my score if these concerns—especially regarding the SFT cold-start issue, modality fairness, and multi-task baselines—are adequately addressed and clarified during the rebuttal.

---

> ### Author Rebuttal · Authors · 2026-03-26
>
> > 1. The catastrophic failure of standard GRPO on specific tasks... suggests a cold-start issue due to a lack of domain knowledge in the base model, rather than an algorithmic failing... I am concerned that HARPO’s relative gains might disappear with proper SFT.
>
> We thank the reviewer for this thoughtful suggestion, and we agree that the role of SFT in our study is worth clarifying. HARPO and GRPO are compared using the same base instruct model without SFT warmup. (1) This follows established GRPO studies  (i.e. Dr GRPO, DeepSeek R1), which argues against SFT warm-up, by highlighting its impediment on LLM reasoning capabilities. (2) Similar evaluation protocols (i.e. comparing reasoning-RL without SFT) are frequently accepted in prior work (QoQ-Med/DRPO, Open-Reasoner-Zero), and we follow this to ensure adequate evaluation. (3) HARPO and GRPO are initialized with the same on-policy setup, where advantages are largely estimated from the previous policy; HARPO differs primarily by rescaling advantages. Given the same value estimation technique, we expect the cold-start setting to affect both methods similarly. While SFT warmup is a well taken point, we kindly appeal to the reviewer to consider the above points, especially given that retraining models is difficult to accommodate within the tight review timeframe.
>
> > 2. Why restrict its (HARPO's) framing and evaluation strictly to social behaviors when it contains zero social priors? Have you evaluated HARPO on standard, non-social multi-task reasoning benchmarks...?
>
> We thank the reviewer for the suggestion and will clarify the framing in the introduction and conclusion. (1) While HARPO does not encode explicit social biases, it is motivated by a key property of social behavioral tasks (heterogeneous reward structures), as reflected in the Human Behavior Atlas paper. (2) We wish to demonstrate HARPO’s practical utility, and the social behavior setting provides a practical testbed for evaluating generalization in low-resource behavioral settings. While HARPO's generalization for other heterogeneous settings (i.e. code, math) is undeniably exciting, the motivation for a unified policy in such domains is less clear.
>
> > 3. How exactly were datasets requiring audio... evaluated on pure VL baselines...? Were ASR transcripts provided, or was the audio modality ignored?
>
> We thank the reviewer for this point and will clarify further in Section 4.1. We agree that modality availability is critical, and thus evaluated against vision+audio baselines (e.g., Qwen2.5-Omni, HumanOmni, OmniSapiens-RL), in addition to VL. VL models are included only to broaden coverage given the limited availability of frontier omni models, and transcripts from the datasets are used to retain audio information for VL. Importantly, comparing HARPO vs. RL baselines also use the same base vision + audio model, which also ensures differences arise from the RL algorithms rather than modality gaps.
>
> > 4. How does HARPO compare against other recent LLM-adapted multi-task gradient balancing techniques...? An empirical or theoretical comparison against these concurrent methods is necessary to prove HARPO’s superiority...
>
> We thank the reviewer for this suggestion. We added EMAGRPO as an additional expt, as this seems most relevant to multi-task setting (other methods, i.e. CTWA, emphasize multi-objective within the same task). HARPO exhibits improved performance compared to EMAGRPO on most tasks, with a better avg. performance rank (HARPO 2.10 vs EMAGRPO 2.78).
>
> | Models | EMO | HUM | INT | PTSD | ANX | DEP | SEN | SAR | SOC | NVC | Avg. Rank ↓ |
> |--------|-----|-----|-----|------|-----|-----|-----|-----|-----|-----|--------------|
> | EMAGRPO | 73.36 | 63.50 | 52.62 | 98.39 | 90.68 | 73.36 | 68.28 | 77.75 | 30.58 | 15.70 | 2.78 |
> | HARPO | 76.55 | 69.85 | 50.52 | 98.39 | 91.98 | 78.87 | 77.61 | 70.64 | 25.40 | 14.54 | **2.10** |
>
> > Finally, regarding the social evaluation itself, it would be a nice addition to briefly include or discuss recent causal and social video datasets like Read the Room and CausalChaos! to further strengthen the video reasoning assessment.
>
> We thank the reviewer for the suggestion and will include a brief discussion of Read the Room and CausalChaos, highlighting their focus on socially grounded video understanding and complementary task formulations. At the same time, we note that Human Behavior Atlas contains IntentQA and Social-IQ for social reasoning. Nonetheless, we will include these additional datasets to better contextualize our evaluation and clarify the scope of our claims.
>
> > Other points
>
> Responded to in our above comments.
>
> >  I am very open to raising my score if these concerns... are adequately addressed and clarified during the rebuttal.
>
> We are keen on improving the paper according to feedback, and have provided clearer details of our experimental rationales. We are open to including more experiments to improve our paper, if the reviewer still deems necessary.

---

> > ### Author Rebuttal · Reviewer_SL1F · 2026-04-04
> >
> > I appreciate the authors' detailed reply and would like to see these clarifications reflected in the revised version. However, since my main concerns are only partially resolved, I will keep my current score unchanged.

---

> > > ### Author Response · Authors · 2026-04-04
> > >
> > > **Inclusion of Additional Experiment Addressing Final Outstanding Reviewer Concern: SFT Cold-Start Initialization**
> > >
> > > ---
> > > > I am very open to raising my score if these concerns—especially regarding the SFT cold-start issue, modality fairness, and multi-task baselines—are adequately addressed and clarified during the rebuttal.
> > >
> > > > However, since my main concerns are only partially resolved, I will keep my current score unchanged.
> > >
> > > We sincerely thank the reviewer for taking the time to consider our rebuttal and for acknowledging our clarifications. In our earlier response, we provided an additional baseline experiment (EMAGRPO) and clarified the comparison against vision-language models, including the use of transcripts for audio to ensure modality fairness.
> > >
> > > At this stage, based on the reviewer's initial comments, we understand that the primary remaining concern likely relates to the absence of a cold-start SFT phase prior to applying GRPO and HARPO. We fully appreciate the importance of this point, and would like to respectfully share an experimental update that we hope helps to address this concern. Over the past few days following our initial rebuttal, we made a concerted effort to secure additional computational resources in order to run new experiments incorporating a cold-start SFT (using 10% of the Human Behavior Atlas training data) for both GRPO and HARPO. We present the full results (with dataset-level breakdowns) in the table below. Across the majority of datasets in the Human Behavior Atlas benchmark, we observe that HARPO continues to outperform GRPO, even when both methods are initialized with this 10% SFT warm-up.
> > >
> > > We hope that this additional experiment helps to more fully address the reviewer’s concerns regarding the SFT initialization. As this appears to have been the primary remaining point of uncertainty, and given the reviewer’s openness to reconsider the evaluation in light of additional experimental evidence, we would be truly grateful if these new results could be taken into careful consideration. We sincerely hope that this clarification helps to strengthen confidence in our findings as part of the final assessment of our paper.
> > >
> > > | Model  | chsimsv2 | cremad | daicwoz | meld_emo | meld_sen | mmpsy_anx | mmpsy_dep | mmsd | mosei_emo | mosei_sen | ptsd | tess | urfunny | intentqa | mimeqa |
> > > |--------|----------|--------|----------|-----------|-----------|------------|------------|------|------------|------------|------|------|----------|----------|--------|
> > > | HARPO  | 0.766    | 0.527  | 0.548    | 0.612     | 0.681     | 0.864      | 0.866      | 0.575| 0.663      | 0.656      | 0.916| 0.886| 0.351    | 0.494    | 0.129  |
> > > | GRPO   | 0.843    | 0.620  | 0.302    | 0.559     | 0.618     | 0.786      | 0.805      | 0.285| 0.639      | 0.621      | 0.916| 0.993| 0.299    | 0.456    | 0.122  |
> > > | Diff   | -0.076   | -0.093 | 0.246    | 0.053     | 0.062     | 0.078      | 0.061      | 0.290| 0.023      | 0.035      | 0.000| -0.107| 0.052   | 0.039    | 0.007  |
> > > | Winner | GRPO     | GRPO   | HARPO    | HARPO     | HARPO     | HARPO      | HARPO      | HARPO| HARPO      | HARPO      | TIE  | GRPO | HARPO    | HARPO    | HARPO  |

---

### Official Review · Reviewer_iqxX · 2026-03-15

**Soundness:** 3
**Presentation:** 3
**Significance:** 3
**Originality:** 3
**Overall Recommendation:** 4
**Confidence:** 3

**Summary:**

This paper proposes a simple yet effective improvement to GRPO for reinforcement learning training. The key idea is to consider the contribution of each sample and task during policy optimization by dynamically computing advantage weights for each trajectory. These weights regulate how different samples contribute to the policy update, helping training remain balanced across diverse tasks. The method is evaluated on social behavior processing tasks, where experiments demonstrate improved performance compared with existing methods.

**Compliance With Llm Reviewing Policy:**

Affirmed.

**Key Questions For Authors:**

Please see comments in the Strengths And Weaknesses section.

**Limitations:**

Please see comments in the Strengths And Weaknesses section.

**Strengths And Weaknesses:**

Strengths
- The proposed method is simple and intuitive, requiring only a lightweight modification to the GRPO training procedure.
- The paper conducts extensive experiments in the social behavior processing domain, providing solid empirical validation.
- The proposed approach demonstrates consistent improvements over baselines across the evaluated tasks.

Weaknesses
- The proposed method mainly introduces a reweighting mechanism for advantages during GRPO training. Similar ideas such as advantage normalization or sample reweighting have been explored in prior RL and RLHF literature, such as in [1]. The paper would benefit from a discussion of how the proposed approach relates and differs from them.
- The performance of the GRPO baseline seems unusual. From the examples and plots shown in Figures 1 and 2, the model trained with GRPO appears to produce very limited (one-word) reasoning traces. This behavior seems somewhat unusual compared to typical GRPO-trained models reported in the literature. It would be helpful if the authors could provide additional analysis to explain this phenomenon. For example, reporting the average reasoning length during training could help determine whether the model suffers from reasoning collapse or catastrophic forgetting of reasoning ability.


[1] Hong et al., Adaptive Preference Scaling for Reinforcement Learning with Human Feedback, 2024

---

> ### Author Rebuttal · Authors · 2026-03-26
>
> We would like to thank Reviewer iqxX for taking the time to read our paper and providing useful feedback. We are eager to continue improving the paper according to the feedback provided, and have appended our responses for the reviewer's perusal.
>
> > The proposed method mainly introduces a reweighting mechanism for advantages during GRPO training. Similar ideas such as advantage normalization or sample reweighting have been explored in prior RL and RLHF literature, such as in [1]. The paper would benefit from a discussion of how the proposed approach relates and differs from them
>
> We thank the reviewer for the useful suggestion. Indeed, the paper can benefit from an extended discussion on how HARPO differs from advantage normalization or sample reweighting. To this end, we will include the following discussion, as an extended portion of the related work in section 2. Specifically, while prior methods (e.g., advantage normalization or sample reweighting in RE++/ RLOO) apply static, per-batch scaling to stabilize gradients and perform advantage normalization, HARPO introduces a structured and dynamic balancing mechanism. HARPO explicitly tracks contribution signals over time and across tasks and explicitly rebalances according to a geometric mean. This enables the method to estimate how much each task or sample contributes to optimization, and adjust accordingly. Second, HARPO performs hierarchical and multiplicative modulation of advantages, with centering and inertial smoothing, ensuring that updates are balanced without distorting training stability. This stands in contrast to conventional normalization (i.e. RE++), which does not explicitly enforce balanced contribution across heterogeneous tasks in an adaptive and structured way. We will add this further discussion into the main text, and we sincerely hope that this addresses the reviewer’s concerns.
>
> > The performance of the GRPO baseline seems unusual. From the examples and plots shown in Figures 1 and 2, the model trained with GRPO appears to produce very limited (one-word) reasoning traces. This behavior seems somewhat unusual compared to typical GRPO-trained models reported in the literature. It would be helpful if the authors could provide additional analysis to explain this phenomenon. For example, reporting the average reasoning length during training could help determine whether the model suffers from reasoning collapse or catastrophic forgetting of reasoning ability
>
> We thank the reviewer for the helpful suggestion. Indeed, we agree that providing additional analysis to explain this phenomenon could help to clarify. Accordingly, we will add the following analysis to Section 4.5. We conduct a targeted analysis of the evolution of relative task advantages under GRPO, focusing on tasks such as SAR and HUM. We observe that their relative advantages progressively diminish over training. When examined alongside the heterogeneity of the advantage distributions (Appendix Figure 7, highlighting differences in sparsity), this suggests that the effect is driven by imbalanced reward and advantage signals across tasks. In particular, tasks with sparse or low-magnitude advantages are increasingly underrepresented in gradient updates, while tasks with denser or higher-variance advantages (i.e. INT) appear to dominate optimization. This behavior is consistent with established observations in policy gradient methods, where advantage normalization or group-based scaling can bias updates toward high-variance or high-magnitude signals [3, 2, 1], leading to under-optimization of weaker signals. Our analysis therefore provides a mechanistic explanation for the observed degradation on SAR and HUM, and highlights the need for explicitly regulating heterogeneous advantage contributions. Sincerely, we hope that this additional analysis can address the reviewer’s comments.
>
> **References:**
>
> [1] Wei Dai, Peilin Chen, Chanakya Ekbote, and Paul Pu Liang. Qoq-med: Building multimodal clinical foundation models with domain-aware grpo training. arXiv preprint arXiv:2506.00711, 2025.
>
> [2] Yuki Ichihara, Yuu Jinnai, Tetsuro Morimura, Mitsuki Sakamoto, Ryota Mitsuhashi, and Eiji Uchibe. Mo-grpo: Mitigating reward hacking of group relative policy optimization on multi-objective problems. arXiv preprint arXiv:2509.22047, 2025.
>
> [3] Shih-Yang Liu, Xin Dong, Ximing Lu, Shizhe Diao, Peter Belcak, Mingjie Liu, Min-Hung Chen, Hongxu Yin, Yu-Chiang Frank Wang, Kwang-Ting Cheng, et al. Gdpo: Group reward-decoupled normalization policy optimization for multi-reward rl optimization. arXiv preprint arXiv:2601.05242, 2026.

---

> > ### Author Rebuttal · Reviewer_iqxX · 2026-04-02
> >
> > Thanks to the authors for the clarification. The authors have provided additional discussion in response to the initial concerns, which appears reasonable and partially addresses the issues raised. However, further experimental results would strengthen the analysis. For instance, it would be valuable to demonstrate whether the performance improvement of the proposed method over the base GRPO method increases as the imbalance in reward and advantage signals across tasks becomes more severe.

---

> > > ### Author Response · Authors · 2026-04-03
> > >
> > > We sincerely thank Reviewer iqxX for this helpful suggestion. We agree that further insight into HARPO’s mechanism is valuable. We added an empirical analysis comparing HARPO and GRPO on a weakly performing task (SAR). Accordingly, we observe that after step 350, SAR exhibits diminished advantages relative to other tasks, reflected by a lower relative task advantage (defined as the ratio between a task’s mean advantage magnitude and the average mean advantage magnitudes across all tasks). Under GRPO, this imbalance is not corrected, and SAR’s reward correspondingly declines in later training stages. In contrast, HARPO adaptively rescales the advantages of underrepresented tasks, which is reflected in a higher mean advantage magnitude after step 350, and a corresponding sustained increase in SAR reward. This trend is consistently observed in the relative task advantage and reward curves during training, where HARPO maintains higher SAR reward and relative task advantages compared to GRPO as training progresses, ultimately leading to a better performance on the SAR task (70.64 for HARPO vs 53.58 for GRPO). These results empirically demonstrate that HARPO mitigates imbalance in advantage distributions, particularly when certain tasks experience disproportionately weakened learning signals.
> > >
> > > We thank the reviewer for the opportunity to clarify this mechanism, and hope this addresses their concern.

---

### Official Review · Reviewer_2Mhz · 2026-03-16

**Soundness:** 3
**Presentation:** 3
**Significance:** 3
**Originality:** 3
**Overall Recommendation:** 3
**Confidence:** 3

**Summary:**

The paper introduces HARPO, an RL method that extends GRPO for heterogeneous multitask training. The main idea is to modulate advantages at both the sample and task levels, which rebalances gradient contributions across tasks. The authors apply it to train OmniSapiens-7B 2.0, a multimodal model for social behavior processing across multiple tasks.

**Compliance With Llm Reviewing Policy:**

Affirmed.

**Key Questions For Authors:**

1. Can the authors explain how many independent training runs were conducted? Also, how sensitive is HARPO to the KL coefficient $\beta$ and the EMA parameter $\beta_p$? In practice, what would you suggest is a good value for these hyper-parameters?
2. OmniSapiens-7B 2.0 underperforms on SOC. What causes the loss in performance? Does the HARPO mechanism sacrifice performance on certain tasks to improve others?

**Limitations:**

My main concern is the absence of variance reporting. Without knowing if results are from a single run, the reliability of the claims cannot be assessed. A sensitivity analysis for key hyperparameters ($\beta$, $\beta_p$) is also missing.

**Strengths And Weaknesses:**

## Strengths

1. HARPO achieves good average rank across tasks as compared to RLOO, GRPO, and RE++, with performance gains in multitask settings.
2. The geometric-mean-based modulation is a simple and principled way to incorporate task-level information into the advantage function.
3. The evaluation covers 10 tasks spanning text, audio, and video modalities.

## Weaknesses

1. The notation $\tilde{A}$ (the clipped surrogate) is not clearly explained and I didn't find the formula for $\tilde{A}$ in the text. Also, the use of capital $A$ for a scalar advantage is confusing since capital letters usually denote matrices. The paper should clarify or use lowercase.
2. The numerical section reports single scalar values for all metrics without standard deviations or confidence intervals, and the number of independent training runs is not specified. Moreover, results are averaged across a task's associated datasets but its not clear to me whether this accounts for dataset-level or run-level variation. Without this information, it is not easy to assess the reported gains.
3. The HARPO objective includes a KL penalty $\beta D_{\mathrm{KL}}(\pi_\theta \| \pi_{\mathrm{ref}})$ but its not clear how $\beta$ or the EMA smoothing parameter $\beta_p$ were chosen. It is unclear whether these were tuned per-task or globally.
4. HARPO shows poor performance on Social Reasoning (SOC) and this 5-point drop is not discussed or explained.

---

> ### Author Rebuttal · Authors · 2026-03-26
>
> > The notation $\tilde{A}$ (the clipped surrogate) is not clearly explained and I didn't find the formula for $\tilde{A}$ in the text. Also, the use of capital $A$ for a scalar advantage is confusing since capital letters usually denote matrices. The paper should clarify or use lowercase.
>
> We thank the reviewer for pointing out this valuable aspect. Indeed, the clipped surrogate can be explained more clearly. We agree to expanding the main text to include the derivation of the clipped surrogate and clarify that the advantage is a scalar and not a matrix.
>
> > The numerical section reports single scalar values for all metrics without standard deviations or confidence intervals, and the number of independent training runs is not specified. Moreover, results are averaged across a task's associated datasets but it's not clear to me whether this accounts for dataset-level or run-level variation. Without this information, it is not easy to assess the reported gains.
>
> We thank the reviewer for this important point, and agree to enhance clarity. We will change the main text to include: (1) The mean and standard deviation computed across datasets within each task. Please find this previewed table below. This will describe performance variation across datasets, following practices in foundational benchmarks (HELM). (2) We will also clarify in the main text, that we ensure robust evaluation by following standard protocols in established RL research (i.e. DPO, GRPO). This involves balancing the high computational cost of RL by reporting a single random independent training run per model, and thereafter validating the result robustness with additional empirical robustness analysis (ablations, mechanism studies). Our method validation strictly follows these protocols: First, we conduct ablations (Section 4.5) over all key components of the proposed method, isolating to validate that performance gains arise from intended design. Second, we perform analyses to verify that the mechanism behaves as expected (e.g., studying the weakly performing GRPO task in Section 4.5), to ensure that improvements are not unintentional, but due to core HARPO mechanisms. We will additionally clarify this in our paper.
>
> | Models | EMO | HUM | INT | PTSD | ANX | DEP | SEN | SAR | SOC | NVC |
> |--------|-----|-----|-----|------|-----|-----|-----|-----|-----|-----|
> | **OmniSapiens-7B 2.0 (ours)** | **76.55 ± 17.86** | **69.85** | **50.52** | **98.39** | **91.98** | **78.87 ± 8.01** | **77.61 ± 10.20** | **70.64** | 25.40 | **14.54** |
>
> > The HARPO objective includes a KL penalty but its not clear how beta or the EMA smoothing parameter were chosen. It is unclear whether these were tuned per-task or globally.
>
> We thank the reviewer for raising this helpful comment. Based on the reviewer's feedback, we wish to make several improvements to our manuscript. (1) The EMA smoothing parameter is applied globally across all tasks and is not tuned per task, as HARPO is designed for unified multi-task optimization; we will clarify this in Appendix C.1.
> (2) The chosen EMA coefficient (0.95) follows standard practice in optimization and representation learning studies, where high-momentum averaging (0.9–0.99) is commonly used to stabilize noisy updates (e.g., Adam, Mean Teacher). We adopt 0.95 as a moderate value within this established range and will include this rationale in Appendix C.1.
> (3) We also analyze the effect of EMA through a sensitivity study. In Section 4.5, we compare HARPO without inertial control (EMA = 0) against the full method (EMA = 0.95), and further examine varying EMA values. Results show that larger EMA coefficients reduce stochastic fluctuations in modulation factors and improve training stability. We will make this analysis more explicit in the main text under a dedicated subsection on inertial control and EMA sensitivity.
>
> > HARPO shows poor performance on Social Reasoning (SOC) and this 5-point drop is not discussed or explained.
>
> We thank the reviewer for this useful comment. We will expand section 4.2 to include the following study on the limitations of HARPO for the SOC task. Our empirical analysis suggests that SOC represents a relatively niche form of social reasoning, requiring fine-grained intent inference from sparse cues. Comparing the predictions of HARPO with RLOO (the best-performing baseline on SOC), we observe that HARPO captures the correct broad social or affective direction, while RLOO produces more interaction-specific predictions (e.g., “happy” vs. “very happy and proud”). This indicates that RLOO may better specialize in SOC-style pragmatic reasoning, whereas HARPO’s balanced optimization prioritizes broader cross-task performance. We view this trade-off as an important direction for future work: preserving generalization across diverse social behavior tasks without sacrificing performance on nicher capabilities.
>
> > Other key questions
>
> Responded in the above comments.

---

> > ### Author Rebuttal · Reviewer_2Mhz · 2026-04-04
> >
> > I thank the authors for the rebuttal. However, I will maintain my current score. I appreciate the clarification that the reported variance represents dataset-level variation, but some tasks exhibit very high variance (e.g., $\pm17.86$ for EMO) which makes it difficult to assess the reliability of the reported gains against baselines. On a different note, the clarifications on notation, the EMA parameter, and the SOC task trade-off will improve the paper.

---

> > > ### Author Response · Authors · 2026-04-04
> > >
> > > >  I appreciate the clarification that the reported variance represents dataset-level variation, but some tasks exhibit very high variance (e.g.,  for EMO) which makes it difficult to assess the reliability of the reported gains against baselines.
> > >
> > > We thank the reviewer for this point. We are concerned that there may be some ambiguity in how the reported variance is interpreted, and we would like to clarify this to ensure the results are assessed appropriately, especially prior to final evaluation of our paper. We sincerely hope that the following clarifications can address the reviewer's concern:
> > >
> > > 1. **Dataset versus method variability:**
> > >    The reported variance reflects dataset-level variability, which is not inherent to our method, and is consistently observed across all baselines. It does not indicate instability specific to our method, nor does it diminish HARPO's gains. To this end, applying the same variance reporting to the baseline methods RE++, GPG, GRPO, and RLOO yields similar variance ranges (e.g., EMO: ±16.64–21.24; DEP: ±7.39–10.58; SEN: ±6.53–37.46), highlighting that this variability is a property of the datasets rather than the training algorithms.
> > >
> > > 2. **HARPO possesses comparable variance as baselines, but improved mean performance:**
> > >    Given that all methods exhibit similar dataset-level variability (because some datasets are more challenging than others), differences in performance are most appropriately assessed at the aggregated mean level. We therefore report mean performance across multiple datasets, which provides a more stable and representative estimate of general task performance as opposed to performance on a single dataset. Under this setting, HARPO consistently achieves higher mean performance with comparable variance as the baselines, indicating that the observed gains are systematic across the datasets present. In fact, this interpretation (claiming performance gains based on mean metrics aggregated across heterogeneous datasets) is widely regarded as an established and robust empirical evaluation practice (i.e. GLUE [2], SuperGLUE [1]).
> > >
> > > 3. **Limited scope of affected tasks:**
> > >    Regardless of points 1 and 2, these multiple dataset settings arise only in a subset of tasks (EMO, DEP, SEN), which constitute a small portion of the overall benchmark. The majority of tasks do not exhibit such dataset-level variability.
> > >
> > > 4. **Alignment with our core claim:**
> > >    Our central claim is not based on improvements within a single task (EMO), but on consistent cross-task performance (i.e. across the 10+ tasks on Human Behavior Atlas) under heterogeneous conditions. HARPO and OmniSapiens 2.0 are designed to achieve balanced learning across diverse behavioral tasks. Accordingly, our conclusions are supported by consistent improvements across multiple tasks and overall ranking trends, rather than isolated gains on any single task (i.e. EMO); and as mentioned in 3., most of the tasks in this benchmark do not encounter dataset-level variability.
> > >
> > > 5. **Additional bootstrapping statistical experiment highlights low within-dataset variance.** To further address concerns the reviewer concerns, we conduct an additional bootstrap analysis to quantify variance, reporting per-dataset confidence intervals (CIs) using 10,000 resamples, following standard NLP/ML practice for distribution-free uncertainty estimation (An empirical investigation of statistical significance, Berg et al., 2012) . The resulting per-dataset CIs are considerably small. For instance, within the EMO task, individual dataset CIs range from ±0.0044 to ±0.0075, highlighting that within dataset performance estimates are highly consistent. This suggests the variance noted is attributable to cross-dataset aggregation over the four EMO datasets (Points 1–2), rather than to within-dataset instability or method unreliability.
> > >
> > > Taken together, we believe that the observed variance does not diminish the validity or reliability of our results, as it does not affect the cross-task improvements that underpin our main contributions, nor does it reflect a significant weakness of our method. We thank the reviewer once again, and sincerely appeal for consideration of the clarifications made, prior to the final assessment.
> > >
> > > ---
> > > **References:**
> > >
> > > [1] Alex Wang, Yada Pruksachatkun, Nikita Nangia, Amanpreet Singh, Julian Michael, Felix Hill, Omer Levy, and Samuel Bowman. Superglue: A stickier benchmark for general-purpose language understanding systems. Advances in neural information processing systems, 32, 2019.
> > >
> > > [2] Alex Wang, Amanpreet Singh, Julian Michael, Felix Hill, Omer Levy, and Samuel Bowman. Glue: A multi-task benchmark and analysis platform for natural language understanding. In Proceedings of the 2018 EMNLP workshop BlackboxNLP: Analyzing and interpreting neural networks for NLP, pages 353–355, 2018

---

### Decision · Program_Chairs · 2026-04-30

**Decision:**

Accept (regular)

**Comment:**

I think the paper makes a valuable contribution by providing an RL method that extends GRPO for heterogeneous multitask training. Some reviewers raised some concerns regarding the experimental evaluation and the SFT cold-start issue. However, I think that the authors provided enough evidence on the performance of the proposed algorithm with their constructive rebuttal.